# Emergence of Grounded, Optimally Compositional Spatial Language among Homogeneous Agents

## Abstract

A mechanism of effective communication is integral to human existence. An essential aspect of a functional communication scheme among a rational human population involves an efficient, unambiguous, adaptive, and coherent apparatus to convey one's goal to others. Such an effective macro characteristic can emerge in a finite population through incremental learning via trial and error at the individual (micro) level, with nearly consistent individual learning faculty and experience across the population. In this paper, we study minimal yet pertinent aspects of glossogenetics, specifically primal human communication mechanisms, through computational modeling. In particular, we model the process as a language game within the fabric of a decentralized, multi-agent deep reinforcement learning setting, where the agents with local learning and neural cognitive faculties interact through a series of dialogues. Our model seeks to achieve the principle of least effort and overcome the poverty of stimulus among homogeneous agents through mirror networks. In our examinations, we observe the emergence of successful and efficient communication among static and dynamic agent populations through consistent learning.

## 1 Introduction

Effective communication via signals is the key to success in a cooperative world, where the goal is to complete the desired tasks by efficiently coordinating among themselves. A functional communication language should be nearly unambiguous, efficient, easily acquirable (culturally transmitted) and rooted in the environment. Language Chomsky (2006); Montague et al. (1970); de Saussure (2011) is an autonomous, culturally transmitted, complex adaptive system realised through multiple modalities - either vocal-auditory or manual-visual which translate mental representations which are internal structures to utterances that represent the surface structure. In many scenarios, a combination of these modalities is applied to express the context unambiguously, which is primarily attributed to the complexity of the context and the environment. In cooperative AI and cognitive science, language games Wittgenstein (1954); David (1969); Arrington (1954); Steels (1997; 2003); Wagner et al. (2003) which was motivated by the picture theory of language Wittgenstein (1954) and operant conditioning theory Skinner (1986) are empirical computational models developed to study the origin, evolution, and acquisition of human languages. The game setting involves a bottom-up simulation model which usually consists of multiple artificial agents (neural or non-neural) equipped with sufficient cognitive abilities and sometimes sensory-motor systems interacting in a shared environment through vocal or non-vocal means and subsequently learning from the outcomes of the interactions. The language structures that emerge in these settings are never equivalent to human languages, since human languages are refined through millions of years of cultural evolution. However, language games can provide deep insights into the emergence of various aspects of human language mechanisms, such as syntactic structures Garcia-Casademont & Steels (2016), compositionality, word order, generalization, brevity, stability, statistical regularity, complexity, coherence, and linguistic divergence.

With the recent advancement in the field of deep learning Mnih et al. (2013) with respect to computational tractability, one could observe rigorous applications of deep learning and deep reinforcement learning involving multiple agents in the context of language games Lazaridou & Baroni (2020); Dafoe et al. (2020), especially referential/discrimination games Lazaridou et al. (2017); Havrylov & Titov (2017), reconstruction

games Kharitonov et al. (2020), navigation/action games Kajić et al. (2020); Mordatch & Abbeel (2018) and visual communication games Qiu et al. (2022). A few of these focus on the emergence of coherent communication protocols from scratch (tabula rasa) in a multi-deep-agent setup Sukhbaatar et al. (2016); Foerster et al. (2016); Havrylov & Titov (2017); Lazaridou et al. (2017; 2018). A few others target the pertinent linguistic universals of natural languages, such as the symbolic grounding Mordatch & Abbeel (2018); Kottur et al. (2017); Lin et al. (2021), compositionality Mordatch & Abbeel (2018); Kottur et al. (2017); Li & Bowling (2019); Ren et al. (2020); Wang et al. (2016); Andreas (2018), generalization Baroni (2020); Chaabouni et al. (2020), brevity regularity Rita et al. (2020); Kharitonov et al. (2020), the cultural and architectural transmission Dagan et al. (2020); Ren et al. (2020), language structures through ease-of-teaching pressure Li & Bowling (2019) and networked communication Gupta et al. (2020). Some of the recent works also provide deeper analysis pertaining to the nature and factors affecting the semiotic dynamics underlying the emergence of language and language constructs. Kottur et al. (2017); Resnick et al. (2020); Tucker et al. (2022) delve into the factors and constraints such as selectionist criteria, utility, informativeness, memory capacity, and learning capabilities that contribute to the development of compositionality and Graesser et al. (2019); Eccles et al. (2019); Gaya et al. (2016) analyze conditions, inductive biases and intrinsic motivation required for the emergence of a coherent language. Another direction in which language emergence is being evaluated is along the dimension of scale Chaabouni et al. (2022); Rita et al. (2021), where the correlation between language characteristics and system complexity, and population dynamics is examined, while Lazaridou et al. (2020) incorporates pre-trained general language models to develop task-specific language models. Choi et al. (2018) explores the obverter technique Batali (1998) which explores emergent communication among pseudo-homogeneuos agents, where the speaker is assumed to be always true, while the listener calibrates its parameters to align itself with the speaker. Chaabouni et al. (2019) studies the existence of inverse correlation of word length and input frequency which exists in natural human language.

### 1.1 Our Contribution

In this paper, we study the emergence of certain coherent properties of a language along with other key factors effecting the language among a multi-agent population. We develop a game setup allowing significant complexity compared to the existing Lewis signalling games settings in terms of combinatorial possibility of mapping the words to a concept by listener for the novel words uttered by the speaker. Additionally, we explored the notion of interchangeability property in the language which enables agents to simultaneously synchronise the bidirectional mappings along with their active role (either listening or speaking) in the game which ensures continuity and internal consistency. In this paper, we also introduce efficient communication through the principle of least effort, where the agents are encouraged to convey information in a way that minimizes complexity or cognitive effort. Moreover, we study the emergent macro behaviour which materializes through the micro dynamics involving all the above functionalities.

## 2 Problem Formulation

In this paper, our objective is to enable the emergence of coherent symbolic structures among a population of deep neural agents through decentralized learning and self-organization in language games. Our setting consists of $N$ deep neural agents populated on a graph world $G = (V, E)$ which is embedded on a bounded 2D plane (flat earth) where $V$ is the set of vertices and $E$ is the set of edges. All the nodes are similar in shape. However, they possess two relevant features, location and color which distinguish them from each other. The location is unique for a node, although they can have the same color. Our agents are homogeneous in nature, where they can perform both comprehension and reproduction of language. This property is referred to as homogeneity/interchangeability which is one of the core properties of human language. Our language game (illustrated in Figure 1) is as follows:

1. At each instant in the game, one agent is paired against another to initiate a semiotic cycle of dialogue consisting of $D$ conversations. In dialogue, one random agent takes the role of speaker, while the other agent is the listener (step ① in Figure 1).

2. In a conversation, the speaker agent chooses a target node (the topic of conversation) uniformly at random from the world (unknown to the listener) and it will try to communicate the target node to the other agent by presenting the utterance using an appropriate conceptualization and vocal language on a noise-free, face-to-face, discrete channel where everything said is heard (steps ② to ⑤).

3. The listener attempts is to decipher the meaning of the utterance and correctly identify the target node and thus accept the utterance by providing evidence of understanding. The interaction is subsequently rewarded according to the interpretation outcome which is shared among the participating agents (steps ⑥ to ⑧).

4. In case of failed communication, the speaker discloses the target node to the listener (step ⑨). Learning occurs through the induction of hypotheses (the innate linguistic structure that is characterized by neural networks) based on payoffs, and disclosures.

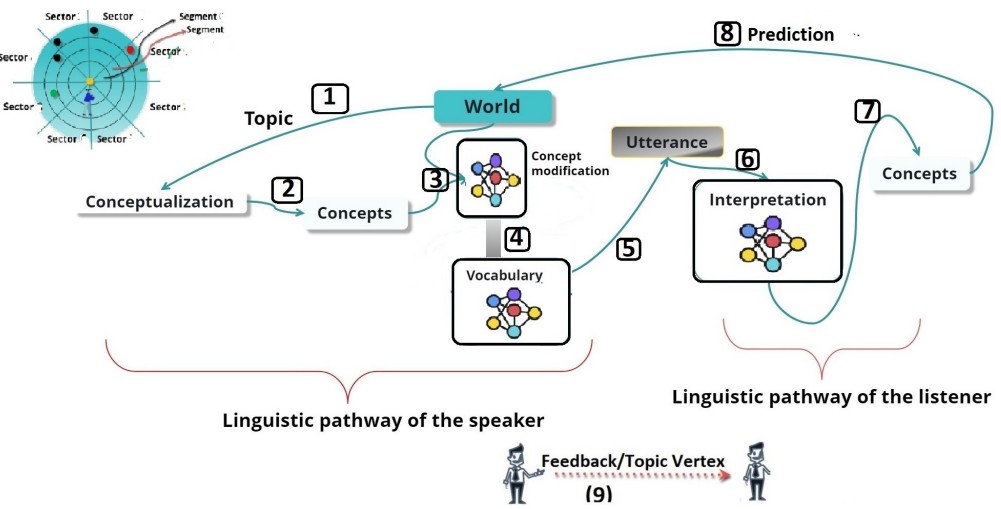

**Figure 1:** Semiotic pathway illustration of the guessing game

We formulate our setting using a multi-agent Markov game framework Littman (1994); Puterman (2014) since we aim for the emergence of symbolic structures through interactions among agents who possess the cognitive ability to extract and reinforce commonalities across multiple experiences. Here, we assume that the agents only have a partial observation of the environment, which aligns with real human scenarios where one can only be aware of his local surroundings and perceive the world in a coarse form. The state of the environment at time step $t$ is denoted by $s^{(t)} \in \mathcal{S}$, where $\mathcal{S}$ is the set of all the environment states. We let $o_i^{(t)} \in \mathcal{O}$ be the partial observation of agent $i$, which is characterized by the function $f_i : \mathcal{S} \mapsto \mathcal{O}$, where $\mathcal{O}$ is the set of all possible observations. At time instant $t$, agent $i$ chooses a random action $a_i^{(t)}$ which is dependent on the current observation according to a parameterized stochastic policy $\pi_{\theta_i}(\cdot|o_i^{(t)})$ which is a conditional probability mass function over $\mathcal{A}$ conditioned on the observation $o_i^{(t)}$. For agent $i$, each state transition yields a random reward $r_i^{(t)}$ according to the function $\mathcal{R} : \mathcal{S} \times \mathcal{A} \times \mathcal{S} \mapsto \mathbb{R}$. The system evolution is stochastic in nature and characterized by the probability transition function $\mathcal{P} : \mathcal{S} \times \mathcal{A} \times \mathcal{S} \mapsto [0, 1]$, where $\mathcal{P}(s, a, s') = \Pr(s^{(t+1)} = s'|s^{(t)} = s, a^{(t)} = a)$ which is the conditional probability of next state is $s'$ conditioned on the current state and action being $s$ and $a$ respectively. The collective goal of the agent population is to collaboratively seek a policy $\pi_{\theta*} = [\pi_{\theta_1^\star}, \pi_{\theta_2^\star}, \ldots, \pi_{\theta_N^\star}]$ that maximizes the globally averaged long-term return over the network based solely on local information, $i.e.$,

$$\theta_i^\star = \arg\max_{\theta \in \Theta} J_i(\theta), \text{ with } J_i(\theta) = \mathbb{E}_{\pi_{\boldsymbol{\theta}}, \mu} \left[ \sum_{t=0}^{T-1} \mathbf{r}_i^{(t)} \right]. \tag{1}$$

where $\mathbb{E}_{\pi_\theta, \mu}[\cdot]$ is the expectation with respect to all $T$ length trajectories generated using the stochastic policy $\pi_\theta$ with initial distribution $\mu$ and $\Theta \subset \mathbb{R}^s v$ is a compact and convex set.

# 3 Domain Ontology

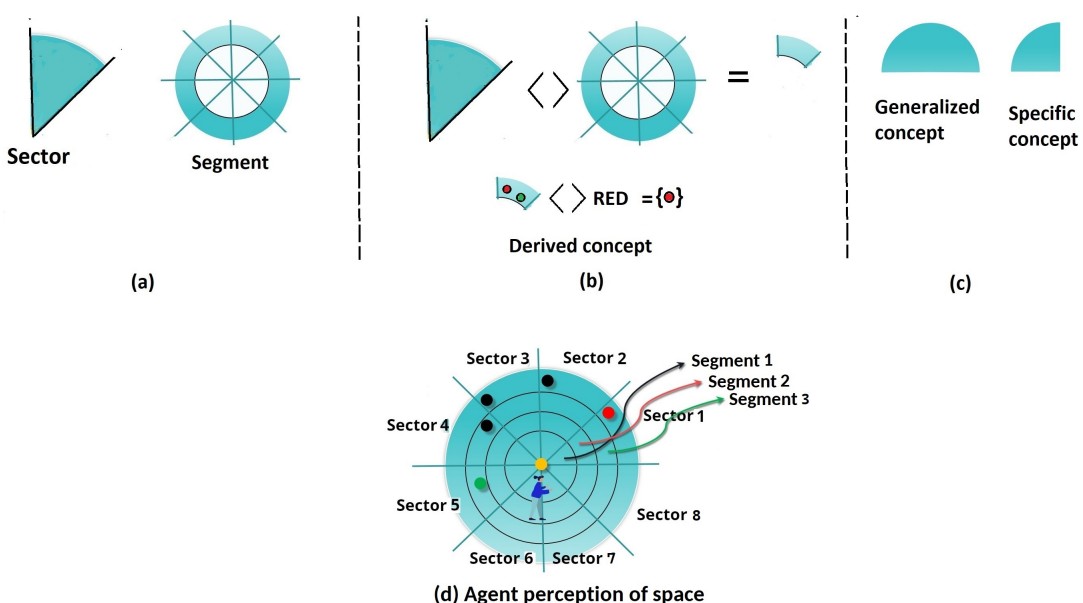

**Figure 2:** Agent Ontology

The ontology Guarino & Giaretta (1995); Mark et al. (2003) of the agent is the concept space $\mathcal{C} = \mathcal{H} \cup \mathcal{W} \cup \mathcal{B} \cup \{\bot\}$ which consists of a finite collection of segments $\mathcal{H}$, sectors $\mathcal{W}$, colors $\mathcal{B}$ and the NULL concept $\bot$. We let $\bar{\mathcal{H}} = \mathcal{H} \cup \{\bot\}$, $\bar{\mathcal{W}} = \mathcal{W} \cup \{\bot\}$ and $\bar{\mathcal{B}} = \mathcal{B} \cup \{\bot\}$. A segment is a strip of region in the 2D plane encompassed by outer and inner concentric circles (Figure 2 (a)) centered at a certain point. A sector is defined to be a part of a disc made of the arc of the disc centered at a certain point along with its two radii extending to the boundary of the world. These are spatial deixis which are generally perceived relative to the location of the central point. The segments and sectors provide a conceptualization of space that is grounded in the sensory and physical interactions of the agents with the world and one can relate it to the concepts of cardinal directions in the human discourse. In our setting, the space is conceptualized a priori as discrete and categorical. Each node possesses the intensive property of color and we assume that the agents possess the sensory mechanism to capture the hue range of colors. Hence, we also consider colors as concepts. Roughly, $\mathcal{C}$ represents the hierarchical deep structure of concepts (semantic entities) where one can be either specific (finer) or general (coarser), or disjoint than the other (Figure 2 (c)). The concept space $\mathcal{C}$ is equipped with an operation $<\cdot, \cdot>: \mathcal{C} \times \mathcal{C} \to \mathcal{C}'$, where $\mathcal{C}'$ is the set of derived concepts, which are concepts which can be derived from the basis concepts Andreas (2018); Montague et al. (1970). In our setting, the operation $<>$ is set intersection since our concept space consists of regions and colors. Hence it is both commutative and associative. An illustration is provided in Figure 2 (b). We also maintain a pred-defined injective encoder $\Gamma : \mathcal{C} \to \mathbb{Z}$ which maps the abstract basis concepts in $\mathcal{C}$ to discrete integers. Given any topic node, the agent can conceptualize the vertex in terms of the tuple $<\texttt{segment}, \texttt{sector}, \texttt{color}> \in \bar{\mathcal{H}} \times \bar{\mathcal{W}} \times \bar{\mathcal{B}}$ relative to the current location of the agent. It's important to note that there is an abuse of notation in this representation, as $<>$ typically denotes a binary operation, and in this case, it should be interpreted as $<\texttt{segment}, <\texttt{sector}, \texttt{color}>>$. For a given vertex $u \in V$, we consider the function $\mathcal{C}_u : V \to 2^{\bar{\mathcal{H}} \times \bar{\mathcal{W}} \times \bar{\mathcal{B}}}$ which maps vertices to their corresponding conceptualizations relative to the source vertex $u$. For a given (source, topic) vertex pair $(u, x)$, one can have more than one conceptualization possible, i.e., $\mathcal{C}_u(x) \subseteq \bar{\mathcal{H}} \times \bar{\mathcal{W}} \times \bar{\mathcal{B}}$. Hence our setting can be categorized as "*guessing game*" (Section 1.3.2 of Steels (2012)). The complexity of the guessing game is substantially high due to the inherent meaning ambiguity arising from the existence of more than one possible distinct concept for a given unknown message. Meaning uncertainty arises because multiple concepts can

possibly be associated to a novel word and the listener cannot, with only one exposure, determine which meaning is intended by the speaker. The available information for learning or understanding is limited or insufficient for the listener to comprehend. This complicates the construction of a shared vocabulary, as aligning meanings through communication becomes more challenging. This is Quine's "Gavagai" problem also referred to as *Poverty of stimulus* Quine (1960).

**Assumption:** In this paper, we assume that the ontology possessed by all the agents is commensurable and they all conform to the same ontological framework to avoid inconsistent perspectives and thus evade the Tower of Babel situation Iliadis (2019); Mark et al. (2003). Also, we assume that each agent possesses an episodic memory to hold the entire sequence.

## 4 Grounded Vocabulary Learning

The lexis ($\Psi$) of a language is a finite catalog of all $q$-letter words available a priori to an agent. A vocabulary bidirectionally maps lexis (phonological entities) to meanings (semantic entities) where one is able to evoke the other Ren et al. (2020). This symbolic association is referred to as the property of groundedness. For a population of agents to successfully communicate, there should exist a shared, coherent vocabulary among the population. This implies that the vocabulary possessed by the agents should hold the same meaning for everyone to successfully communicate verbally among themselves. Ideally, the mapping should be isomorphic. Apparently, in every realistic scenario, this is not the case, which transpires into various language characteristics like homonyms and synonyms. Initially, there is no ex-ante meaning associated with the words, and hence no coherence among the agents exists and we aim to foster common grounding among agents incrementally, which is fully shaped by past linguistic experience. This is referred to as the symbol grounding problem Steels (2012). We achieve this through verbal interactions between them, where they extract and reinforce similarities across multiple episodes incrementally through evidence of understanding which can be either positive or negative. This trial and error based calibration process shapes, reshapes, and enforces the mental mapping, where the phonological expressions become more efficient and established through repeated use Bisk et al. (2020); Arrington (1954), and eventually drives the system to a dissipative structure Prigogine (1987) which enables common ground for expressing concepts.

**Defintion (Emergent vocabulary):** An emergent vocabulary $\mathcal{M}$ is a shared mapping (not necessarily bijective) function between lexis $\Psi$ and the concept space $\mathcal{C}$, *i.e.*, $\mathcal{M} : \Psi \leftrightarrow \mathcal{C}$ collectively agreed upon by all the agents in the population de Saussure (2011). Note that there are $|\mathcal{C}|^{|\Psi|}$ possible vocabularies for all the agents to agree upon, which makes it unlikely for all agents to converge on the same vocabulary without some mechanism for coordination and consensus.

**Definition (Compositionality):** A languange is compositional Andreas (2018); Montague et al. (1970) if the utterance of each derived concept is determined by the utterances of its basis concepts. Formally, for the derived concept $c =< g, h, q >$, we have $\mathcal{M}(c) = \mathcal{M}(g)\mathcal{M}(h)\mathcal{M}(q)$, with the implicit operation of concatenation connecting them.

**Assumption:** During each dialogue, the source and target vertices corresponding to each conversation are chosen uniformly at random.

The probabilistic regular grammar corresponding to the language we consider here is the following:

$$D \rightarrow C_1C_2C_3 \ with \ probability \ 1$$
$$C_1 \rightarrow h, \ where \ h \in \mathcal{M}(\mathcal{H}) \ with \ probability \ \mu(h|C_1)$$
$$C_1 \rightarrow \epsilon, \ with \ probability \ \mu(\epsilon|C_1)$$
$$C_2 \rightarrow w, \ where \ w \in \mathcal{M}(\mathcal{W}) \ with \ probability \ \mu(w|C_2)$$
$$C_2 \rightarrow \epsilon, \ with \ probability \ \mu(\epsilon|C_2)$$
$$C_3 \rightarrow b, \ where \ b \in \mathcal{M}(\mathcal{B}) \ with \ probability \ \mu(w|C_3)$$
$$C_3 \rightarrow \epsilon, \ with \ probability \ \mu(\epsilon|C_3)$$

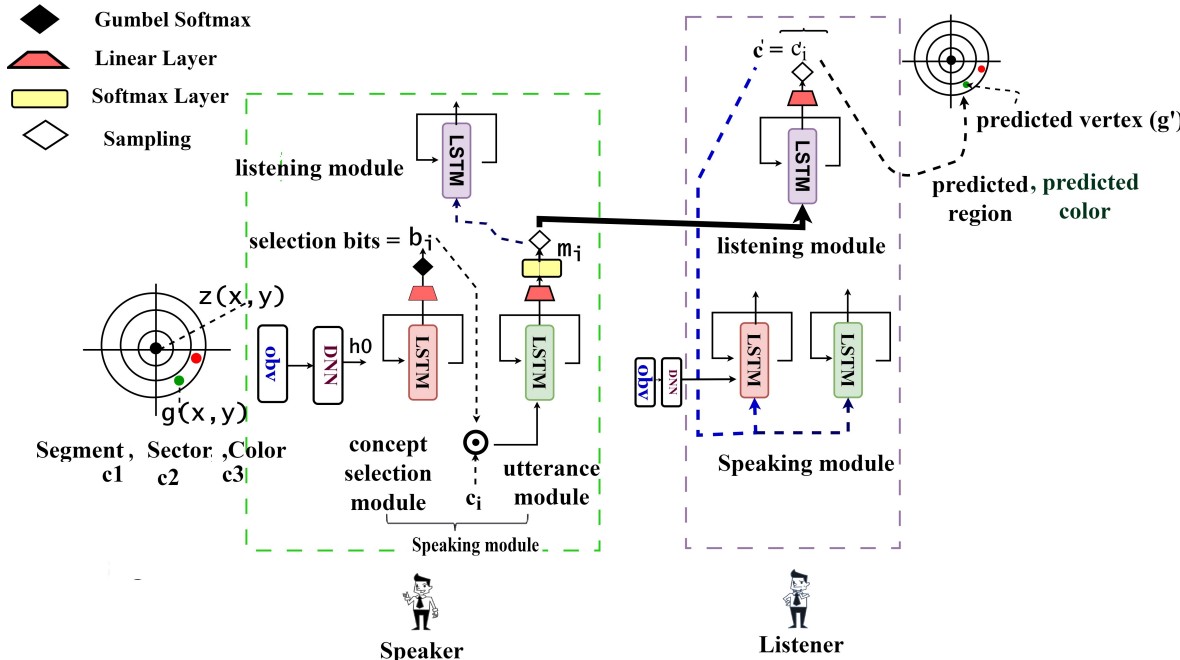

**Figure 3:** Policy architecture of the semiotic pathway

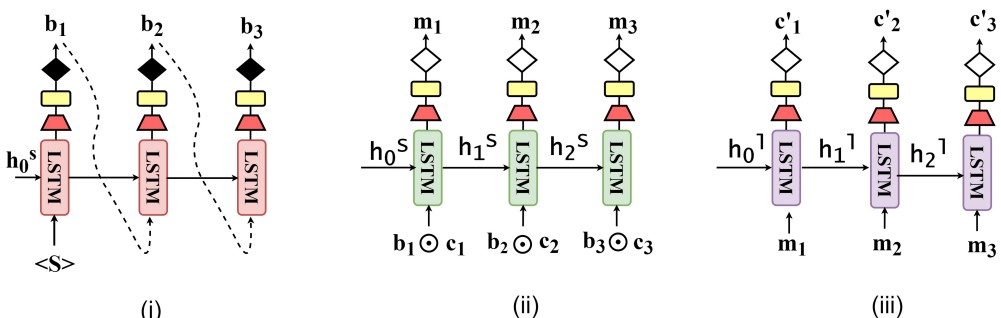

**Figure 4:** Unfolded view of the respective LSTMs.Best viewed in color

The language is finite and regular and hence learnable in the limit (Gold sense, Theorem 2.6 of Niyogi (2006)) under the above assumption. The speaker exhibits an inherent categorical bias, to structure messages in a specific order ($\mathcal{M}(\texttt{segment}), \mathcal{M}(\texttt{sector}), \mathcal{M}(\texttt{color})$) reflecting the agent's conceptualization process. However, the listener module does not share this bias and instead treats each word as potentially belonging to any category. This discrepancy in how the speaker and listener process information makes it more challenging to establish a shared language, as the listener does not rely on the same categorical ordering as the speaker.

## 4.1 Policy Architecture

The policy architecture of the agent is modeled using stochastic neural networks. Each agent consists of two modules: speaking (concept-selection and utterance) and listening. All the modules are implemented using RNN (recurrent neural networks) to allow for continuous and sequential communication. Here $\theta, \psi$, and $\phi$ represent the parameters of the utterance network, concept-selection network, and listening network respectively. All the modules of listener and speaker have to synchronize through trial and error for a successful communication language to emerge. In our setting, we perform decentralized learning with decentralized execution Foerster et al. (2016). Our agents are independent learners Tan (1993) and the channel between speaker and listener is non-differentiable, which implies that the back-propagation of the listener

does not transmit the gradient backward to the speaker. In our 2D environment, there are $N$ agents and $M$ vertices. The state $\mathcal{S}$ of the game set consists of all relevant details that define the environment. The state of the environment at time $t$ is given by $\mathbf{s}_t = \left[ \mathbf{x}^{(1),\dots,(N)}, \mathbf{z}_t^{(1),\dots,(N)}, \mathbf{q}^{(1),\dots,(N)}, \mathbf{u}_t^{(1),\dots,(N)} \right]^{\top} \in \mathcal{S}$, where $\mathbf{x}^{(i)} \in \mathbb{R}^2$ is the location of the $i^{th}$ vertex in the world, $\mathbf{z}^{(i)} \in \{1, 2, \dots, N\}$ is the current location of agent $i$, $\mathbf{q}^{(i)} \in \mathbb{R}$ is the color of vertex $i$ and $\mathbf{u}_t^{(i)}$ is the utterance in the conversation involving agent $i$. The agent $i$ locally perceives the environment which characterizes the observation vector of the speaker agent $\mathbf{o}_t^{(i)} \stackrel{f_i}{=} \left[ \mathbf{z}_t^{(i)}, \mathbf{g}_t^{(i)}, \mathbf{u}_t^{(i)}, \mathbf{q}^{(\mathbf{g}_t^{(i)})}, \mathtt{DNN}(\mathbf{d}^{(1),\dots,(M)} + \epsilon_d, \mathbf{w}^{(1),\dots,(M)} + \epsilon_w) \right]^{\top}$, where $\epsilon_d \sim \mathcal{N}(0,1)$ and $\epsilon_w \sim \mathcal{N}(0,1)$ are white Gaussian noises, $\mathbf{g}_t^{(i)} \in \{1, 2 \dots M\}$ is the topic vertex, and $\mathbf{d}, \mathbf{w}$ represent the distance and the angle of vertices from the speaker's current vertex respectively. Here $\mathtt{DNN}$ represents a deep neural network which embeds the graph relative to the source vertex. The interaction pathway consists of multiple networks across the speaker and listener agents operating sequentially. The concept-selection network $\pi_\psi$ operates in a one-to-many mode, where the initial hidden vector is obtained through a linear transformation of the observation vector $\mathbf{o}_t$, and the output is fed back as input. This network outputs the conception-selection bit-vector $\mathbf{b}_t$ which is then passed through a differentiable channel to the speaking module $\pi_\theta$ (many-many mode) along with the spatial description $\mathbf{c}_t$ of the topic vertex as $\mathbf{c}_t \odot \mathbf{b}_t$, where $\odot$ is co-ordinate-wise vector product. The network utters the message $\mathbf{m}_t$ which is transmitted to the listener through a non-differentiable (naive categorical sampling) noise-free channel. We use Gumbel-Softmax Jang et al. (2016); Maddison et al. (2016) based sampling to enable differentiability of concept-selection to utterance channel allowing gradients to flow through the sampling process. The Gumbel-Softmax distribution for a given the parameters $p \in \mathbb{R}^K$ is defined as follows:

$$G(\log p)_k = \frac{\exp((\log p_k + \varepsilon)/\tau)}{\sum_{j=1}^{K} \exp((\log p_j + \varepsilon)/\tau)}, 1 \le k \le K,$$

where $G(\log p)_k$ represents the $k^{th}$ element of the one-hot encoding sample $G$, $\varepsilon \sim Gumbel(0, 1)$, and $\tau \in \mathbb{R}$ is the temperature parameter.

The listening module $\pi_\phi$ in the listener agent operates in a many-to-many mode, which means it processes the words in the generated message $\mathbf{m}_t$ sequentially and generates a probability distribution $\pi_\phi(\cdot|\mathbf{m}_t)$ over the entire concept space $\mathcal{C}$. This distribution represents the agent's interpretation of the message in terms of different concepts within the concept space. This distribution is further used to generate the listener interpretation $\mathbf{c}_t'$ through categorical sampling. The complete architecture of the agents is depicted in Figure 3.

## 5 Performance Measure

The objective function of our language game consists of three components: Regularized communication feedback, description length loss, and mirror loss. Each component plays a specific role in shaping the communication outcome and the characteristics of the emergent language.

### 5.1 Regularized, Guided Communication Feedback

Here, we consider the standard RL objective function (finite horizon cumulative reward) with an entropy regularization term. The regularizer offers a few advantages that are conducive to language games. First, entropy regularization encourages exploration and helps prevent early convergence to sub-optimal policies. Second, the resulting policies can serve as a good initialization for fine-tuning to a more specific behavior. Third, the maximum entropy framework provides a better exploration mechanism for seeking out the best mode in a multimodal reward landscape. In the language game, we follow a stochastic, guided feedback mechanism. During a failed interaction, the speaker plausibly guides the listener by pointing out the topic vertex to the listener with a probability $\lambda \in [0, 1]$. This implies that the speaker may or may not provide effective guidance with a certain probability. The speaker and listener subsequently reinforce with respect to the spatial concept $\mathbf{c}_t'$ corresponding to the plausibly communicated topic vertex. This implies that during

the interaction between the speaker $A$ and listener $B$, the interpreted concept $\mathbf{c}'_t$ is taken as

$$\mathbf{c}'_t \sim \lambda \pi_{\phi^B}(\cdot|\mathbf{m}_t) + (1-\lambda)\delta_{\mathbf{c}_t}, \text{ where } \lambda \in [0,1] \text{ and for } E \subseteq \mathbb{R}^k, \delta_x(E) = \begin{cases} 1 \text{ if } x \in E, \\ 0 \text{ othewise.} \end{cases} \quad (2)$$

Here $\delta_x$ is the Dirac measure at $x$ which is a singular measure that places all its probability mass at the single point $x$. In the case of effective guidance, a full reward is associated with the interaction.

## 5.2 Principle of Least Effort

According to the principle of least effort Zipf (2016); Cancho & Solé (2003), language evolves because speakers of the language tend to simplify their speech in various ways in order to obtain a trade-off between understanding and effort. When deciding how to express themselves in a language, speakers consider both their present and future communication needs. This drives the speakers to consider linguistic constructs that are effective in meeting their communication goals and efficient in optimizing their labour. A similar hypothesis connecting the overarching fairness between cognitive load and language exposition is the principle of the economy of thought Mach (1898). It suggests that the human mind, with its limited cognitive resources, seeks to represent the infinite complexities of the world in a way that is efficient and economical. From these arguments, we believe that languages tend to evolve in ways that promote the economy of least thought and linguistic effort where the language users communicate using sentences that are relatively easy to produce and comprehend. Hence, in the post-transient phase of language evolution, sentence length tends to decrease Futrell et al. (2015).

## 5.3 Mirror Networks

To enable homogenity/interchangeability of the language, we consider mirror networks. A mirror neuron Di Pellegrino et al. (1992); Rizzolatti et al. (1996), strictly defined, is a type of neuron that is fired both when the individual executes certain actions and when it observes a strictly or broadly congruent set of actions. In our setting, we want the speaking, listening, and concept selection networks of an agent to be consistent with each other so that the information gained through the comprehension of the language is used for its reproduction and vice versa. Since our networks represent stochastic policies, by consistency, we mean in the Bayesian probabilistic sense (posterior distribution of the parameters conditioned on the message or concept). By ensuring consistency between these policies, you're seeking a coherent, bidirectional relationship between how the agent generates its responses (speaking) and how it interprets and understands incoming information (listening). This implies that the calibration pathway has to update and synchronize all the relevant networks in the direction of consistency. Hence, we consider the following mirror loss:

$$\mathbb{E}\left[ \alpha_1 \underbrace{\mathcal{D}_{KL}\left(\pi_{\theta^A}(\cdot|\mathbf{m}) \,\middle\|\, \pi_{\phi^A}(\cdot|\mathbf{m})\right)}_{\text{speaker congruence}} + \alpha_2 \underbrace{\mathcal{D}_{KL}\left(\pi_{\phi^B}(\cdot|\mathbf{c}') \,\middle\|\, \pi_{\theta^B}(\cdot|\mathbf{c}')\right)}_{\text{listener congruence}} \right.$$
$$\left. + \alpha_3 \underbrace{\mathcal{D}_{KL}\left(\pi_{\phi^B}(\cdot|\mathbf{m}) \,\middle\|\, \pi_{\psi^B}(\cdot|\mathbf{o})\right)}_{\text{concept selection congruence}} \right], \text{ where } \alpha_{1\ldots3} \geq 0,$$

with $\mathcal{D}_{KL}(p_1\|p_2) = \sum_x \log p_1(x) \frac{\log p_1(x)}{\log p_2(x)}$ is the Kullback-Leibler divergence. The above loss is used to quantify the dissimilarity or error between the conditional probabilities of the mirror networks and the corresponding active networks. Minimizing the mirror loss during calibration implies making the mirror networks as similar as possible to the active networks resulting in bidirectional language use.

## 5.4 Poverty of Stimulus

The poverty of stimulus appears in guided feedback scenarios, where the speaker reveals the topic vertex $\mathbf{g}$ (which is the tangible component) to the listener at the end of a failed conversation. However, the conceptualization $C_{\mathbf{z}}(\mathbf{g})$ of the topic vertex $g$ consists of more than one element which makes the novel

message $\mathbf{m}$ of the conversation potentially ambiguous and uncertain. This uncertainty poses a challenge for the listener. To address the meaning uncertainty inherent in the poverty of stimulus situation due to insufficient information, the listener relies on contextual cues, where it distributes the message $\mathbf{m}$ across all the possible conceptualizations $\mathcal{C}_{\mathbf{z}}(\mathbf{g})$ of the topic vertex $\mathbf{g}$ with respect to the source vertex $\mathbf{z}$ and assign different normalized weights or probabilities $w_b$ to each interpretation $b$ based on some predispositions (the factors could be prior knowledge and context):

$$\log \pi_{\phi_B}(\mathbf{c}'|\mathbf{m}) = \frac{\sum_{b \in \mathcal{C}_{\mathbf{z}}(\mathbf{g})} w_b \log \pi_{\phi_B}(b|\mathbf{m})}{\sum_{b \in \mathcal{C}_{\mathbf{z}}(\mathbf{g})} w_b}, \text{ where } w_b \geq 0. \tag{3}$$

Over time, as the listener gains more exposure to the word and its usage in various contexts, the uncertainty decreases, and the listener becomes more adept at determining the intended meaning based on the context of its usage.

## 5.5 Objective function

The performance measure $J(\theta, \psi, \phi)$ of the language game is defined as follows: Let $\mathbb{E}_{\mathcal{I}}[\cdot]$ be the expectation induced by the $r.v.s.$ $\mathbf{m} \sim \pi_{\theta^A}(\cdot|c)$, $\mathbf{c}' \sim \pi_{\phi^B}(\cdot|m)$, $\mathbf{s} \sim \mu$, $\mathbf{o} = f_A(\mathbf{s})$, $\mathbf{o} \to \mathbf{c}$ and $\mathbb{E}_{\mathcal{I}_t}[\cdot]$ be the expectation induced by the $r.v.s.$ $\mathbf{m}_t \sim \pi_{\theta^A}(\cdot|\mathbf{c}_t)$, $\mathbf{b}_t \sim \pi_{\psi^A}(\cdot|\mathbf{o}_t)$, $\mathbf{c}'_t \sim \lambda \, \pi_{\phi^B}(\cdot|\mathbf{m}_t) + (1-\lambda)\delta_{\mathbf{c}_t}$, $\mathbf{s}_t \sim \mu$, $\mathbf{o}_t = f_A(\mathbf{s}_t)$, $\mathbf{o}_t \to \mathbf{c}_t$.

Then $J(\theta, \phi, \psi) = \kappa_1 \mathcal{L}_1(\theta, \phi, \psi) + \kappa_2 \mathcal{L}_2(\theta, \phi, \psi) + \kappa_3 \mathcal{L}_3(\theta, \phi, \psi)$, where $\kappa_1, \kappa_2, \kappa_3 \geq 0$,

$$\mathcal{L}_1(\theta, \phi, \psi) = \underbrace{\mathbb{E}_{\mathcal{I}_t}\left[\sum_{t=0}^{T-1} \mathbf{r}_t + \beta \mathcal{H}(\pi_{\theta^A}(\cdot|\mathbf{c}_t)) + \beta \mathcal{H}(\pi_{\phi^B}(\cdot|\mathbf{o}_t))\right]}_{\text{Regularized cumulative reward}}, \beta \geq 0,$$

$$\underbrace{\mathcal{L}_2(\theta, \phi, \psi) = -\mathbb{E}_{\mathcal{I}}\left[\|\mathbf{b}\|_2^2 + \beta' \mathcal{H}(\pi_{\psi^A}(\cdot|\mathbf{s}))\right]}_{\text{Description length loss (Principle of least effort)}}, \beta' \geq 0$$

$$\mathcal{L}_3(\theta, \phi, \psi) = \mathbb{E}_{\mathcal{I}}\Bigg[\alpha_1 \mathcal{D}_{KL}\left(\pi_{\theta^A}(\cdot|\mathbf{m}) \,\Big\|\, \pi_{\phi^A}(\cdot|\mathbf{m})\right) + \alpha_2 \mathcal{D}_{KL}\left(\pi_{\phi^B}(\cdot|\mathbf{c}') \,\Big\|\, \pi_{\theta^B}(\cdot|\mathbf{c}')\right) +$$

$$\alpha_3 \mathcal{D}_{KL}\left(\pi_{\phi^B}(\cdot|\mathbf{m}) \,\Big\|\, \pi_{\psi^B}(\cdot|\mathbf{o})\right)\Bigg] (\texttt{Mirror loss}),$$

with $\mathcal{H}(\pi(\cdot|s)) = -\sum_a \pi(a \,|\, s) \log \pi(a \,|\, s)$ is the entropy regularizer, where, $\mathcal{H}(\pi(\cdot|s))$ represents the entropy of a policy $\pi$ conditioned on state $s$. The entropy measures the uncertainty or randomness associated with the actions chosen by that policy when in a particular state.

Further, we obtain the gradient of $J$ as follows:

$$\nabla J(\theta, \phi, \psi) = \kappa_1 \nabla \mathcal{L}_1(\theta, \phi, \psi) + \kappa_2 \nabla \mathcal{L}_2(\theta, \phi, \psi) + \kappa_3 \nabla \mathcal{L}_3(\theta, \phi, \psi),$$

where

$$\nabla \mathcal{L}_1(\theta, \phi, \psi) = \mathbb{E}_{\mathcal{I}}\Bigg[(Q_{\mathcal{I}}(\mathbf{s}, \mathbf{m}, \mathbf{b}, \mathbf{c}') - \beta \log \pi_{\theta^A}(\mathbf{m}|\mathbf{c}) - \beta \log \pi_{\phi^B}(\mathbf{c}'|\mathbf{m}) - \beta)(\nabla_{\theta^A} \log \pi_{\theta^A}(\mathbf{m}|\mathbf{c})$$

$$+ \nabla_{\phi^B} \log \pi_{\phi^B}(\mathbf{c}'|\mathbf{m}))\Bigg], \tag{4}$$

$$\nabla \mathcal{L}_2(\theta, \phi, \psi) = \mathbb{E}_{\mathcal{I}}\left[(-\beta' \log \pi_{\psi^A}(\mathbf{b}|\mathbf{o}) - \beta')\nabla_{\psi^A} \log \pi_{\psi^A}(\mathbf{b}|\mathbf{o})\right]$$

$$- \mathbb{E}_{\substack{\mathbf{s} \sim \mu, \\ \mathbf{o}=f_A(\mathbf{s})}}\left[\nabla_{\psi^A} \mathbb{E}_{\mathbf{b} \sim \pi_{\psi^A}(\cdot|\mathbf{o})}\left[\|\mathbf{b}\|_2^2\right]\right] \text{ and}$$

$$\nabla \mathcal{L}_3(\theta, \phi, \psi) = \mathbb{E}_{\substack{\mathcal{I}, \\ \mathbf{c}' \to \mathbf{b}'}}\Bigg[-\alpha_1 \frac{\mathbb{P}(\mathbf{c})}{\mathbb{P}(\mathbf{m})} \nabla_{\phi^A} \log \pi_{\phi^A}(\mathbf{c}|\mathbf{m}) - \alpha_2 \frac{\mathbb{P}(\mathbf{m})}{\mathbb{P}(\mathbf{c}')} \nabla_{\theta^B} \log \pi_{\theta^B}(\mathbf{m}|c')$$

$$- \alpha_3 \frac{\mathbb{P}(\mathbf{b}')}{\mathbb{P}(\mathbf{o})} \nabla_{\psi^B} \log \pi_{\psi^B}(\mathbf{b}'|\mathbf{o})\Bigg], \tag{5}$$

where $Q_{\mathcal{I}}(s, m, b, c') = \mathbb{E}_{\mathcal{I}}\left[\sum_{t=0}^{T-1} \mathbf{r}_i^{(t)} | s, m, b, c'\right]$. Here Equation (4) is obtained by appealing to soft policy gradient theorem Shi et al. (2019) and multi-agent policy gradient theorem Zhang et al. (2018).

### 5.6 Reward Function

We follow a reward mechanism that balances exploration, cooperation, synchronization, accuracy, and efficiency in communication. Agents are trained using a shared reward mechanism by which they learn to cooperate by forming a shared language. To encourage agent exploration, we offer partial and complete rewards, motivating the agent to try different approaches and adapt themselves to make informed decisions during training. Both agents receive a partial reward if the listener infers the right region where the topic vertex is located but fails to identify the topic vertex. This acknowledges the successful transmission of relevant information without complete understanding. A full reward is given if the listener can accurately and unambiguously infer the exact topic vertex from the communicated information. This indicates a high level of successful communication and concept selection. A penalty is given if communication fails in order to discourage the respective concept-vocabulary mapping and to prevent incorrect or ineffective communication choices. The balance between partial and complete rewards, along with the penalty for communication failure, encourages agents to refine their communication strategies over time.

$$\mathbf{r}_t = \begin{cases} \zeta_1 \ (\in \mathbb{R}), \ \text{if} \quad g_t = g'_t, \\ \zeta_2 \ (\in \mathbb{R} \wedge \zeta_2 < \zeta_1), \ \text{if} \quad \mathcal{C}_{z_t}(g_t) \cap \mathcal{C}_{z_t}(g'_t) \neq \emptyset, \\ \zeta_3 \ (\zeta_3 \leq 0 \wedge \zeta_3 < \zeta_2), \ \text{otherwise}. \end{cases} \tag{6}$$

The concept-selection module of the speaker seeks to select the optional spatial description to refer to the topic vertex by deactivating redundant concepts. The mechanism aims to ensure that the sentence corresponding to the generated spatial description is of optimal length to convey the intended meaning effectively. To support optimal word-order selection, we penalize the speaker for choosing a sub-optimal sequence of concepts. In cases where a concept is de-activated, the agent chooses to remain silent at that particular instant of the corresponding generated message. To enable this, the utterance module chooses a "NULL" utterance $\mathcal{C}(\bot)$ to indicate silence. The concept of $\bot$ utterance is significant since we do not explicitly impose it a priori, rather it is learned through interactions. In order to promote consistency and coherence in the use of the $\mathcal{C}(\bot)$ utterance across different word categories in a sentence, we employ a strategy to positively reward $\mathbf{r}'$ the speaker for the reuse of the same word for the $\bot$ irrespective of its temporal position in a sentence. This reward system encourages the emergence of a common word for the $\bot$ across different contexts, regardless of its temporal position in the message $\mathbf{u}_t$.

$$\mathbf{r}'_t = \begin{cases} \zeta'_1 \ (\in \mathbb{R}), \ \text{if} \quad |\{\mathcal{M}(a)|a \in \mathbf{u}_t \wedge a = \bot\}| = 1, \\ \zeta'_2 \ (\zeta'_2 < \zeta'_1), \ \text{otherwise}. \end{cases} \tag{7}$$

## 6 Experiments & Discussion

### 6.1 Experimental Setup

In all the experiments, we consider random initial values for the model parameters. The hyper-parameters (learning rate, batch size, and regularization strength) are fine-tuned through iterative experimentation. The feasible reward values for various scenarios are obtained through an exhaustive, yet rational search. In this paper, we consider a 2D world consisting of a complete graph with 5 vertices whose positions are randomly chosen. There are four agents in this world leading to a total of 12 unique speaker-listener pairs which likely allows for a rich variety of interactions and communication scenarios. Each dialogue consists of 100 conversations. The role switching (speaker to listener and listener to speaker) occurs after every 500 iterations through random selection. During every conversation in a dialogue, a random vertex (except the source vertex) is chosen as the topic vertex. In this paper, we consider two-timescale networks Chung

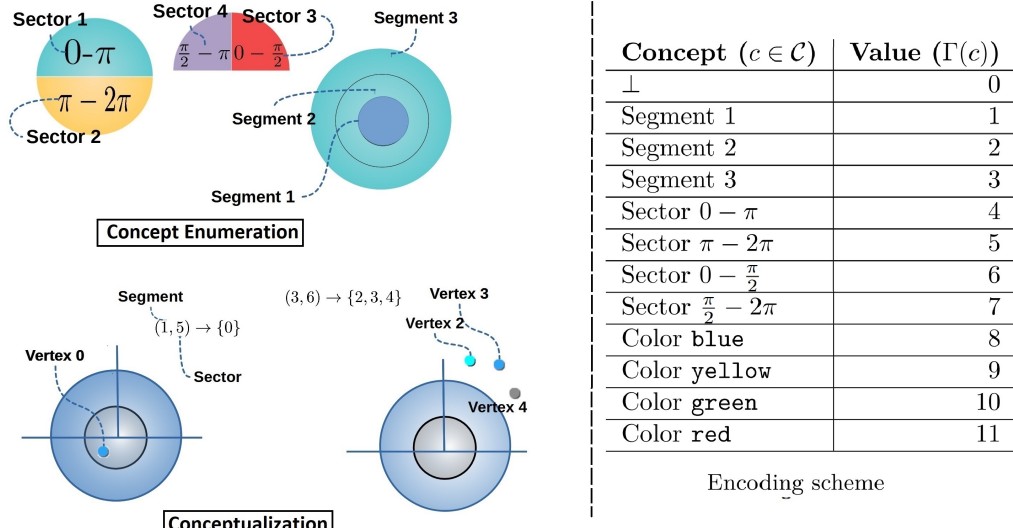

The concept space and the corresponding conceptualization of vertices

| Concept ($c \in \mathcal{C}$) | Value ($\Gamma(c)$) |
|---|---|
| $\perp$ | 0 |
| Segment 1 | 1 |
| Segment 2 | 2 |
| Segment 3 | 3 |
| Sector $0 - \pi$ | 4 |
| Sector $\pi - 2\pi$ | 5 |
| Sector $0 - \frac{\pi}{2}$ | 6 |
| Sector $\frac{\pi}{2} - 2\pi$ | 7 |
| Color blue | 8 |
| Color yellow | 9 |
| Color green | 10 |
| Color red | 11 |

Encoding scheme

**Figure 5:** Concept space and the corresponding conceptualization of vertices

et al. (2018) to obtain synchronized convergence, where the utterance network is calibrated using a faster timescale compared to the conception selection network. In this approach, the concept selection network can be considered to be pseudo-stationary, while the utterance network converges with respect to the stationary values of the concept selection network and this cycle repeats itself in the long run. To achieve this, we employ the vanilla stochastic gradient algorithm with learning rates of the respective networks differing by order of magnitude. This can be formalized as follows: Let $\{e_t\}$ and $\{e'_t\}$ be the learning rates of the concept selection network and utterance network respectively. Then $\{e_t\}$ and $\{e'_t\}$ satisfy the following:

$$e_t, e'_t \in (0,1), \sum_{t \geq 0} e_t = \sum_{t \geq 0} e'_t = \infty, \sum_{t \geq 0} e_t^2 + e'^2_t < \infty, \lim_{t \to \infty} \frac{e_t}{e'_t} = 0. \tag{8}$$

The concept space $\mathcal{C}$ consists of 4 sectors, 3 segments and 4 colors. The concept space $\mathcal{C}$ is illustrated in Figure 5. Since there are overlapping sectors (Sectors 1, 3, and 4) we have a poverty of stimulus situation. The lexis size ($|\Psi|$) is 25. The speaking and listening module within the agent's architecture utilizes an LSTM cell. The observation vector $o_t$ by the speaker agent is transformed into a feature vector ($\in \mathbb{R}^{25}$) by passing it through a fully connected neural network. This feature vector forms the hidden input of the concept selection module ($\psi$) whose hidden size is also taken as 25. The speaking and listening modules are implemented as a single-layer LSTM cell with a hidden size 250. The LSTM networks output the sequence of words or concepts with a maximum length of 3. For the continuous relaxation of categorical distribution within the concept selection module $\tau$ of the Gumbel Softmax, a temperature parameter of 0.5 is utilized. Gradients originating from all modules are clipped with a maximum value of 50. Additionally, successful communication rewards both the speaker and listener with 100, and partial success merits a reward of 50.

The observations emerging from our language game can be summarized as follows:

*We observe that the language reflects the complexity of the environment they describe. The emergent language exhibits an inclination towards minimizing cognitive effort, reflected in the emergent word order, bifurcation of concept space into active and dormant regions and the prevalence of a single-word representation for the $\perp$ concept. This aspect of language can be seen as a reflection of cognitive economy, where speakers aim to convey meaning using the least amount of cognitive effort. It also ties into the broader idea of communication efficiency, where languages evolve to facilitate effective communication while minimizing the cognitive load on speakers and listeners. The frequent convergence of guessing games is guided by the initial exploration in the space of vocabulary-concept mappings during the transient phase, leading to coherence in a finite number*

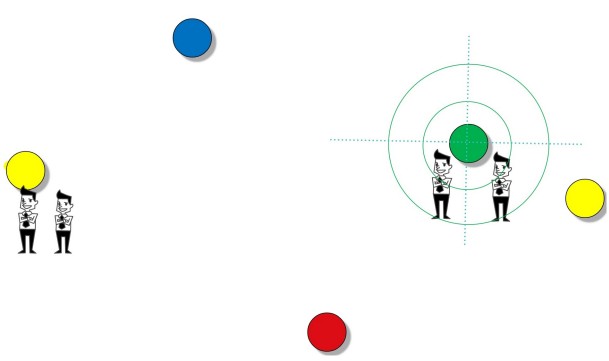

| Source | Conceptualization |
|--------|-------------------|
| 0 | $(3,5) \rightarrow \{1,2,3\}, (3,\perp) \rightarrow \{1,2,3\}, (\perp,5) \rightarrow \{1,2,3\}, (2,5) \rightarrow \{4\}, (2,\perp) \rightarrow \{4\}$ |
| 2 | $(3,7) \rightarrow \{0,3,4\}, (3,\perp) \rightarrow \{0,3,4\}, (\perp,7) \rightarrow \{1,3,4,0\}, (3,4) \rightarrow \{0,3,4\},$ $(\perp,4) \rightarrow \{1,3,4,0\}, (2,7) \rightarrow \{1\}, (2,\perp) \rightarrow \{1\}, (2,4) \rightarrow \{1\}$ |
| 1 | $(3,4) \rightarrow \{0\}, (3,\perp) \rightarrow \{0\}, (\perp,4) \rightarrow \{3,4,0\}, (3,7) \rightarrow \{0\}, (\perp,7) \rightarrow \{4,0\},$ $(2,5) \rightarrow \{2\}, (2,\perp) \rightarrow \{4,2,3\}, (\perp,5) \rightarrow \{2\}, (2,4) \rightarrow \{3,4\}, (2,6) \rightarrow \{3\},$ $(0,6) \rightarrow \{3\}, (2,7) \rightarrow \{4\}$ |
| 3 | $(3,7) \rightarrow \{0\}, (3,\perp) \rightarrow \{2,0\}, (\perp,7) \rightarrow \{0\}, (3,4) \rightarrow \{0\}, (\perp,4) \rightarrow \{0\},$ $(2,5) \rightarrow \{1,4\}, (2,\perp) \rightarrow \{1,4\}, (\perp,5) \rightarrow \{2,4,1\}, (3,5) \rightarrow \{2\}$ |
| 4 | $(2,7) \rightarrow \{0\}, (2,\perp) \rightarrow \{3,0,1\}, (\perp,7) \rightarrow \{0\}, (2,4) \rightarrow \{0,3\}, (\perp,4) \rightarrow \{0,3\},$ $(2,5) \rightarrow \{1\}, (\perp,5) \rightarrow \{2,1\}, (3,5) \rightarrow \{2\}, (3,\perp) \rightarrow \{2\}, (2,6) \rightarrow \{3\}, (0,6) \rightarrow \{3\}$ |

**Figure 6:** The conceptualization of vertices with respect to each source vertex (referred to as source in the table). The syntax followed in the table is as follows: For the source vertex $v \in V$ (source column), the conceptualization column contains $(a,b) \in \Gamma(\mathcal{H}) \times \Gamma(\mathcal{W}) \rightarrow \{u \in V | (a,b) \in \mathcal{C}_v(u)\}$

*of dialogues through reinforcing the successful, yet rare interactions within the dialogues. This coherence, marked by a nearly 100% success ratio during dialogues eventually, signifies the establishment of a robust, shared and grounded language, resilient to variations in agent roles. The synchronization of mirror networks ensures continuity during learning and reproduction, while the emergence of dominating words and adherence to Zipf's law reveal structured emergence, shaping the evolving linguistic landscape. Thus in the intricate tapestry of our decentralized guessing game involving multiple agents, diverse phenomena weave together, depicting the emergence of a shared, grounded, structured and efficient communication system.*

### 6.2 Convergence of Games

In our experiments, we observe that our decentralized guessing games are converging very often. The vocabulary-concept mappings developed by the individual agents during the transient phase are random which enables sufficient exploration to drive the evolution towards coherence in a finite number of dialogues. Coherence implies a more consistent and meaningful use of language, where words and expressions convey clear, grounded and shared meanings. This is corroborated by the convergence of loss functions and the maximization of average reward (average of the rewards of the conversations in a dialogue) as illustrated in Figures 11b and 8b. Note that we maximize the shared cumulative soft rewards by calibrating the policy parameters in the direction of the policy gradient over a non-differentiable channel between the speaker and listener. Positive rewards, attributed to successful communication events, are considered rare. When such events occur, positive rewards are reinforced by policy gradient-based calibration operations. This is further boosted by the guided feedback mechanism, where the listener uses the topic vertex information conveyed by the speaker to calibrate his parameters to match the conceptualization of the topic vertex (possibly ambiguous) with the utterance. These mechanisms help reinforce successful communication strategies. This is observed in most of the trials ($\approx 95\%$), however, in some cases this behaviour is not observed which is primarily attributed to the lack of positive rewards which arises due to random initialization of the neural network weights and the distribution bias of the source, topic vertices pair chosen for the conversations.

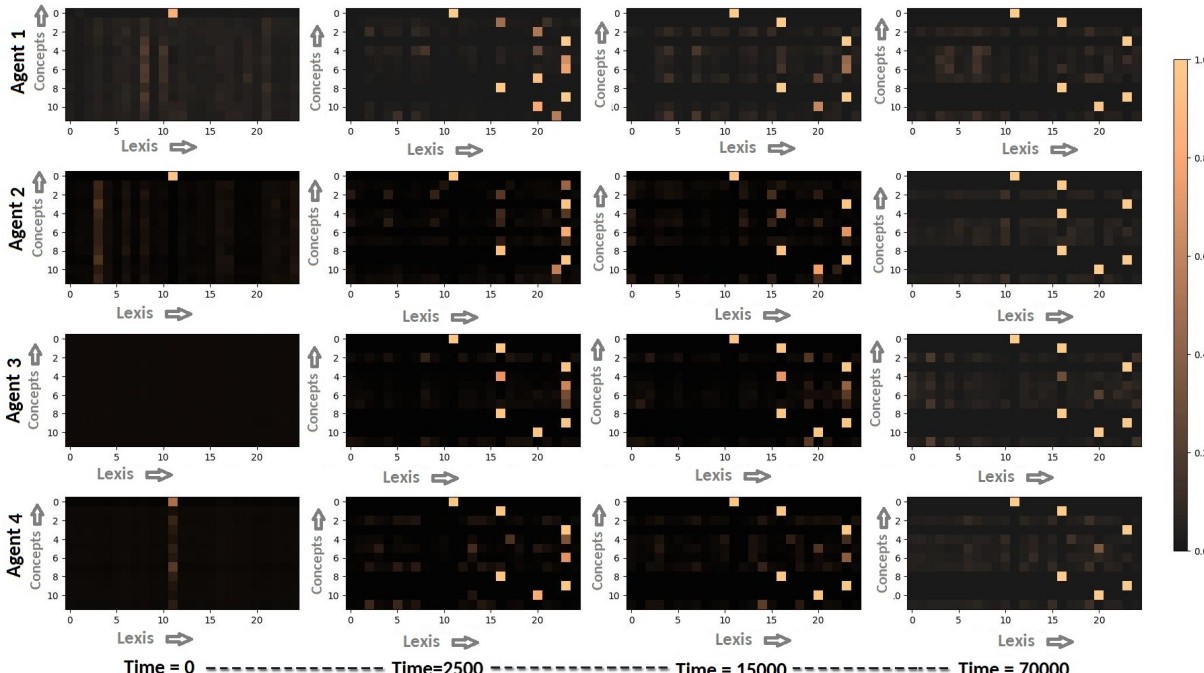

**Figure 7:** Convergence of shared vocabularies among the population is depicted here. Each rectangle represents the vocabulary of an agent, with each row ($k$, where $1 \leq k \leq 4$) showing the evolution of agent $k$'s vocabulary. It is notable that in the later stages of the process, all agents exhibit vocabularies that are nearly identical.

### Guessing game achieves a $100\%$ success ratio over time $\Rightarrow$ Shared language emerges

The success ratio is defined as the frequency of conversations in a dialogue, where the listener is able to identify the topic vertex. The evolution of this success ratio is depicted in Figure 8a over the course of dialogues across the population. This implies that all the dialogue interactions in the conversation among the agent population are successful after a finite number of steps which suggests that the participants are able to achieve their communication objectives (identifying topic vertex) effectively through the medium of language and this is independent of the nature of the agent executing the role of listener and speaker. As the process unfolds, it becomes increasingly apparent that all agents converge towards possessing highly similar vocabularies (Figure 7). The convergence towards near-identical vocabularies likely stems from the agents' interactions and the need for effective communication. Through repeated interactions, agents gradually align their linguistic representations, leading to a shared, grounded lexicon. This shared vocabulary not only facilitates smoother communication but also indicates the agents' ability to adapt and coordinate their language use. In conclusion, the generated language is grounded and mostly unambiguous and the communication model is robust and not heavily dependent on specific characteristics of individual agents.

### 6.3 Cognitive Economy

The agents involved in the language game have demonstrated a remarkable ability to grasp the essential concepts required for effective communication through the application of the principle of least effort. We observe that the order 101 which denotes $<$`segment,` $\perp$`, color`$>$, is the emergent word order (Figure **??**), suggesting a common consensus among the agents that for every topic vertex, communicating two concepts (`segment`, `color`) is adequate (Figure 12). The agents tend to communicate in a way that minimizes cognitive effort. The language tend to adapt to be as efficient as possible, with speakers and listeners preferring forms of expression that require the least amount of cognitive resources. This adaptation is driven by the principle of least effort component of the learning dynamics. To corroborate this claim further, we consider cases where the effect of principle of least effort is reduced significantly (Figure 9). For this purpose, consider the scenarios where $\kappa_2$, representing the weight associated with loss $\mathcal{L}_2$ that quantifies the principle of least

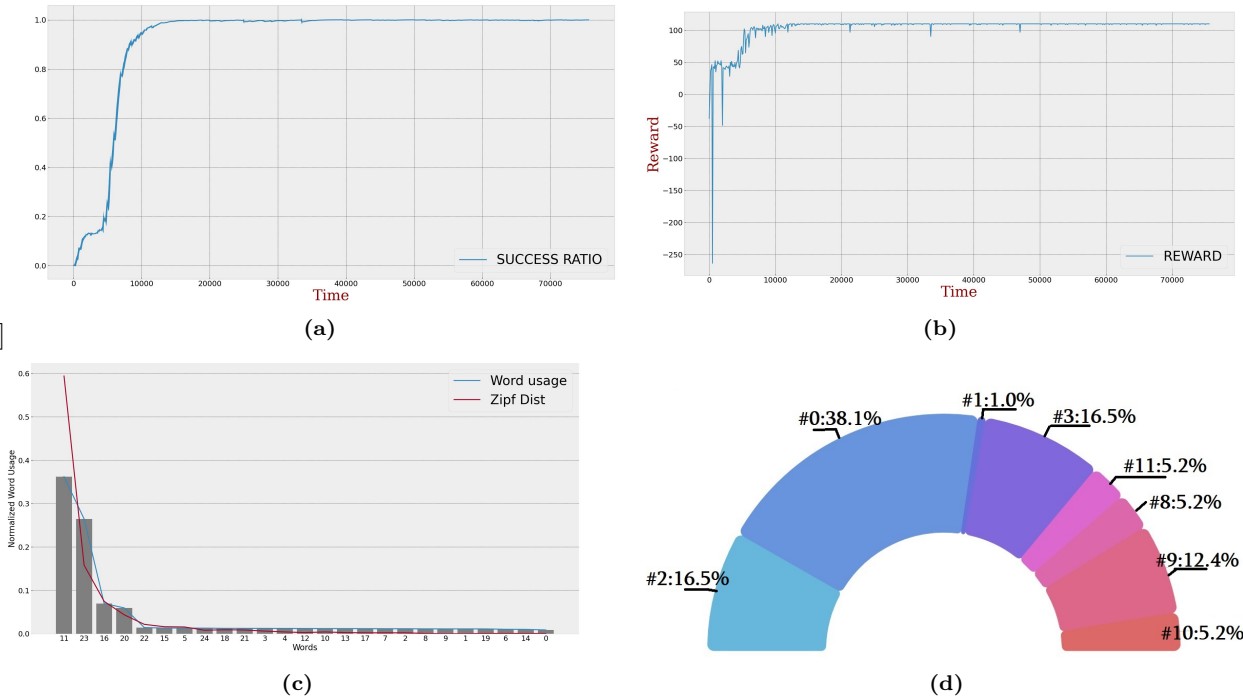

**Figure 8:** (a) Evolution of communication success ratio during dialogues over time (b) Trend of average reward over interactions within a dialogue over time (c) Word usage across all the dialogues over time among the agent population (d) Concept usage across all the dialogues over time among the agent population.

effort, takes on values of 1.1, 0.01, and 0.001 (with $\kappa_1 = 1$ and $\kappa_3 = 1$). For the case when $\kappa_2 = 1.1$, we have minimal conceptualization ( $<$segment, $\perp$, color$>$), of the topic vertex, however for the remaining cases, *i.e.*, $\kappa_2 \in \{0.01, 0.001\}$, we have maximal conceptualization ( $<$segment, sector, color$>$), of the topic vertex which signifies that the principle of least effort is fundamentally attributed to the cognitive economy exhibited by the agents, with higher weights for principle of least effort leading to simpler conceptualizations and lower values prompting more detailed ones.

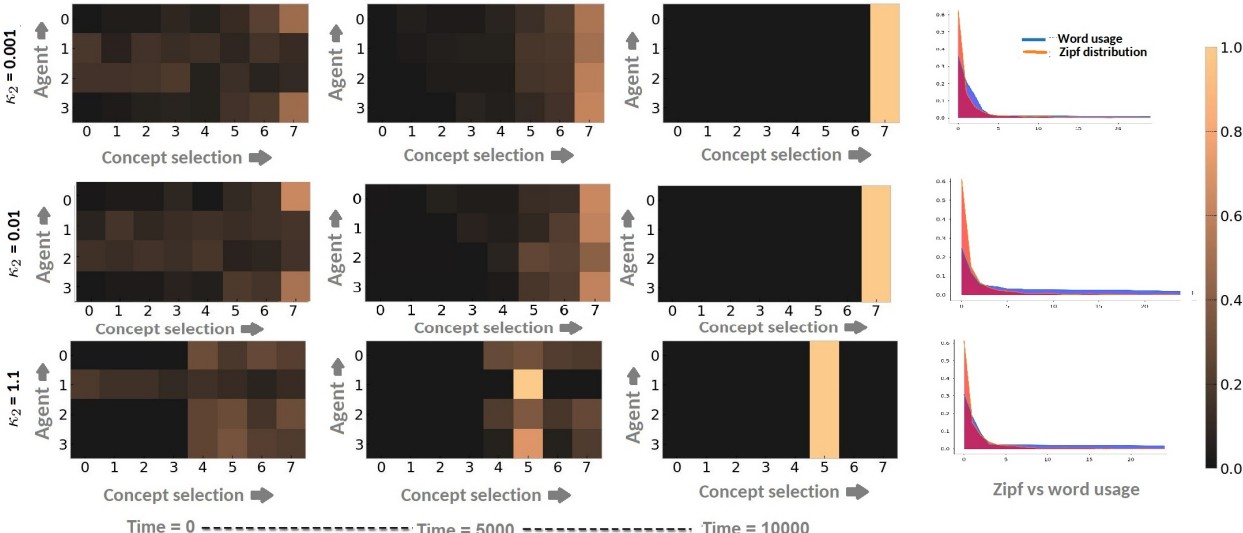

**Figure 9:** Sensitivity of principle of least effort on the cognitive economy

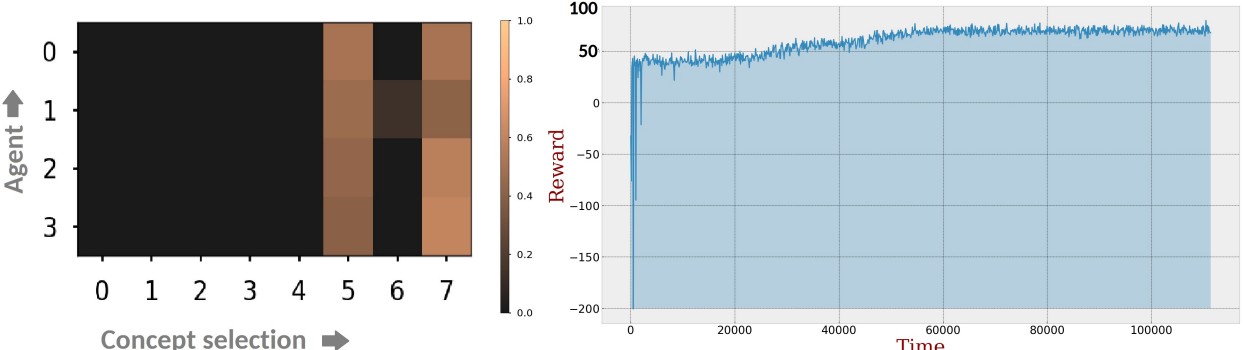

**Figure 10:** In this setting, we have 4 agents with the same static concept space as before residing a graph world with 50 vertices. Note that the world is too complex for the agents to perceive uniquely using their pre-defined concept space. The conceptualization of the topic vertices seems to use as much concepts as possible (60% preferring ``, while 40% preferring ``

To understand the nature of principle of least effort with respect to the complexity of the word, we consider a setting with four agents having the same concept space as before, residing in a graph world of 50 vertices. This world is notably complex, presenting a challenge for the agents to fully comprehend using their predefined concepts. When it comes to conceptualizing the topic vertices, there is an intriguing trend. About 60% of the agents prefer to describe vertices using a combination of segment, sector, and color. On the other hand, the remaining 40% opt for a simpler approach, using segment, a placeholder for silence , and color. This diversity in conceptualization approaches highlights the complexity of the world these agents are trying to understand. Despite their efforts to adapt and communicate effectively, the intricate nature of the graph world poses a significant challenge for the agents in establishing a shared understanding.

### 6.3.1 Bifurcation of Concept Space

Since the topic of conversation is a vertex that is conceptualized using sectors, segments, and colors, a few sectors and segments rarely appear in the conversations due to the distribution of vertices in the 2D world. They remain mostly dormant (inactive) and hence no dominating words for these less-discussed concepts emerge. Contrary to the dormant concepts are the active concepts which are consistently used by the agents to express the topic vertices and hence are alive in the population. This phenomenon suggests that certain concepts may become dominant in communication due to their frequent usage, while others remain less prominent. The above bifurcation of the concept space is highly sensitive to the nature of the graph world and the distribution bias of the source-topic vertex pairs used during interactions. This is illustrated in Figure 8d, where we allude to concepts $4, 5, 6$ and $7$ which are unused and concept 1, which sees minimal usage ($\approx 1.0\%$). Since each color appears once in the world, except for yellow ($\Gamma(\texttt{yellow}) = 9$) which occurs twice and with the word order emerging as 101 ($<$`segment, ⊥, color`$>$), all the `color` concepts are active with `yellow` being more salient in communication.

### 6.3.2 Context Free Expression of Silence

A shared word for expressing silence (`NULL`/⊥) emerges and it is the most popular word. The dynamics of the game settle down to employing a single word to represent ⊥ concept irrespective of the position of ⊥ in the message, a phenomenon vividly illustrated in Figure 8c. As the communication game evolves, a consensus emerges among the agents to use a single word to represent the ⊥ concept. This transformation is additionally accompanied by a noteworthy reduction in the probabilities associated with other words, underscoring the agents' adaptability and the efficiency of their evolving communication system. There is a clear and consistent way of representing the absence of a specific concept in a message. Since 101 ($<$`segment, ⊥, color`$>$) is the globally accepted word order in the population, a ⊥ concept is always present in every conversation which makes its word the most popular word in the process.

### 6.4 Continuity in Language Use (Homogenity/Interchangeability)

In our experiments, we observe that the mirror networks synchronize bidirectional mappings near completely in the agent population. During the interactions, the individual agents calibrate their corresponding mirror networks (listener network for the speaker and utterance network for the listener) with respect to their corresponding active networks (utterance network for the speaker and listener network for the listener) which ensures continuity during role switching. This is achieved by minimizing the mirror loss which converges to 0 as illustrated in Figure 11b. The continuity in learning (utterance comprehension and reproduction) is exemplified in Figure 8a, where one can observe the dips in success ratio (due to role switching) eventually vanish ensuring near continuity. The term "continuity in learning" implies that, over time, the system gets better at maintaining a smooth flow of communication, even when the roles of the agent change from speaker to listener or vice versa. The agents become more proficient at using language through practice and experience, where the information gained during the comprehension of language is applied during its reproduction and vice versa, resulting in effective bidirectional language use.

### 6.5 Emergent Language Characteristics

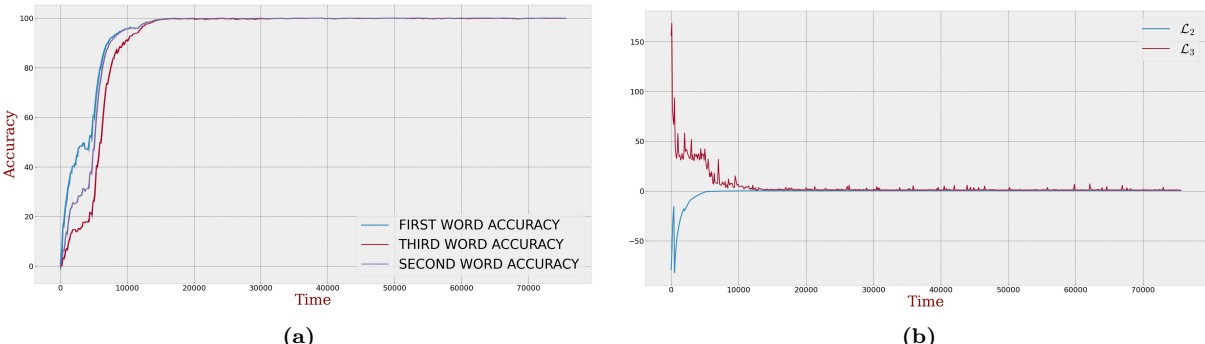

**Figure 11:** (a) First,second and third word accuracy: These metrics measure the frequency of interactions in a dialogue where the listener correctly interprets the first, second, and third words of a sentence, respectively across the population. (b) The evolution of the mirror loss and description length loss across the event population with respect to time

#### 6.5.1 Emergence of Dominating Words

From Figure 7, one can observe that for each active concept ($\{0, 1, 2, 3, 8, 9, 10, 11\}$ from Figure 8d) there exists a dominating word (with probability one) except concept 11 (color `red`). In other words, for each active concept being discussed (except `red`), there is a specific word that is always used when referring to that concept. Further exploring the emergent language, one can observe certain intriguing patterns. For example, the concepts 9 (`yellow`) and 3 (`segment 3`) are represented by the same word 23 (See Figure 7). Since `yellow` is categorized as color and `segment 3` is categorized as segment, the two concepts take different positions in the conceptualization, and hence the presence of the same word at different positions of the message generates different concepts making them unambiguous. Thus the word 23 is a homonym. A similar observation applies to another homonym, word 16, which is associated with concepts 1 (`segment 1`) and 8 (`blue`). It seems that for the `red` vertex, segment information is enough to deduce the vertex and hence color `red` is not learnt. Hence one can observe (Figure 8c), the emergence of four dominating words from the lexis ($|\Psi| = 25$) is mapped to the concept space ($|\mathcal{C}| = 12$). The convergence towards an unambiguous dominating word for each concept could be beneficial in fostering effective communication, as it reduces ambiguity and promotes a shared understanding of the intended meanings behind specific terms.

#### 6.5.2 Emergence of Compositionality

The economy of thought gradually transpires into compositionality in the emergent language since language encapsulates thought. Hence one could observe the emergence of efficient communication strategies, where

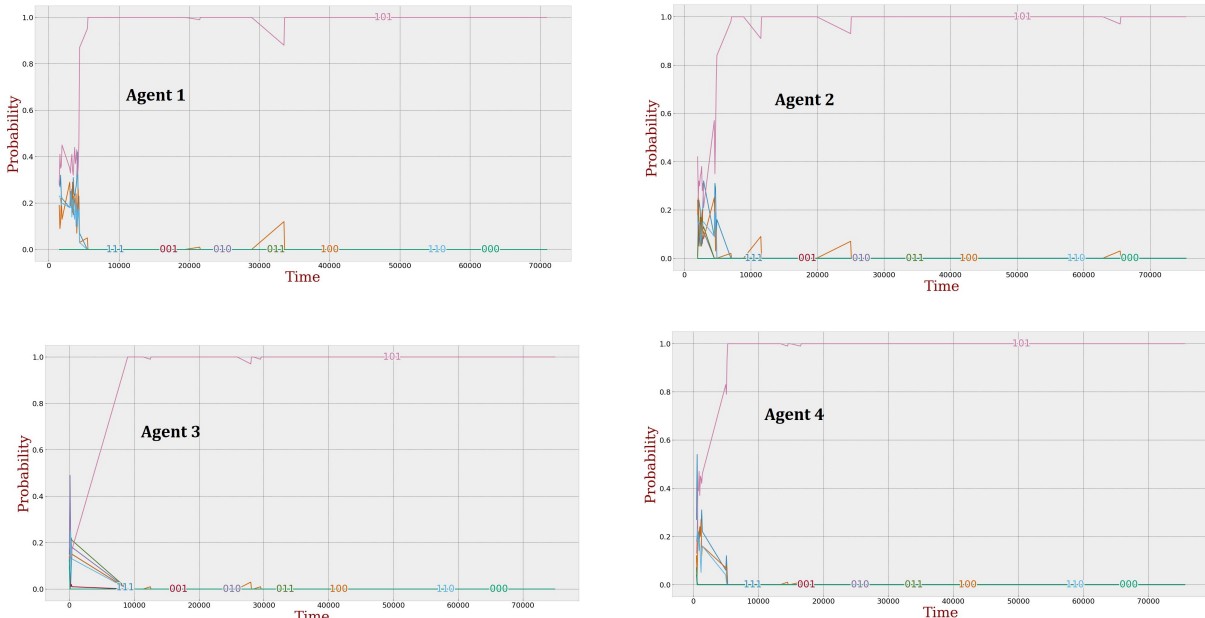

**Figure 12:** Cognitive economy: Evolution of the common word order among the population of agents. Here $y$-axis is the probability of choosing the specific word order. All the agents converge to a common word order 101 which represents $<$`segment`$, \perp,$ `color`$>$

shorter phrases are preferred over longer ones. In our setting, the language that emerged is two word compositional, where it follows a word order which involves `segment` followed by `color`. More complex scenarios can also arise. To illustrate compositionality more vividly, we consider a smaller graph with $N = 3$ vertices (Figure 13) and two homogeneous agents with the concept space as before (Figure 5). The outcomes are depicted in Figures 13 and 14. Notably, no specific word order emerges in this scenario; instead, a blend of word combinations dominates the discourse. It is noteworthy that, in this setup, the number of available colors in the agents' concept space is 4, which is more than the number of vertices. This implies that the agents can communicate the topic vertices by just referring to the color alone. The same holds true for sectors. However, the observed trend reveals that in the majority of conversations (50%), the agents opt for the color alone ($< \perp, \perp,$ `color`$>$), and in 14% of instances, it relies on the sector alone ($<$ `segment`$, \perp, \perp >$). Interestingly, in (20%) of conversations, the agents prefer a combination of segment and color ($<$`segment`, $\perp,$ `color` $>$). The above mixture (dominated by an optimal concept selection) is a sub-optimal limiting behaviour with respect to concept selection and this scenario arises due to the non-convex nature of the objective function operating over a decentralized setting, as shown by the convergence to sub-optimal values in Figure 13). However, the success ratio (Figure 13) Despite this, the success ratio is nearly 100% (Figure 13), and all agents have nearly identical vocabularies (Figure 15), indicating the emergence of an effective semi non-compositional language among the population with 65% of the conversations being non-compositional and 24% two word compositional. Similar sub-optimal behaviors can be expected in human scenarios. Nevertheless, what stands out is that the agents employ one, sometimes two, and rarely three words or stays silent during the dialogue, mirroring patterns observed in human interactions.

### 6.5.3 Adherence to Zipf's Law

Zipf's law is a linguistic phenomenon that describes the distribution of word frequencies within a language. It states that the frequency of any word is inversely proportional to its rank. In other words, the most common word occurs approximately twice as often as the second most common word, three times as often as the third most common word, and so on. We see in the Figure 16 (world setting from Figure 13), the occurrence of the word decreases exponentially for the lower rank words in the language emerged among agents. This aligns with the characteristic pattern described by Zipf's law. To further understand this relationship, we studied

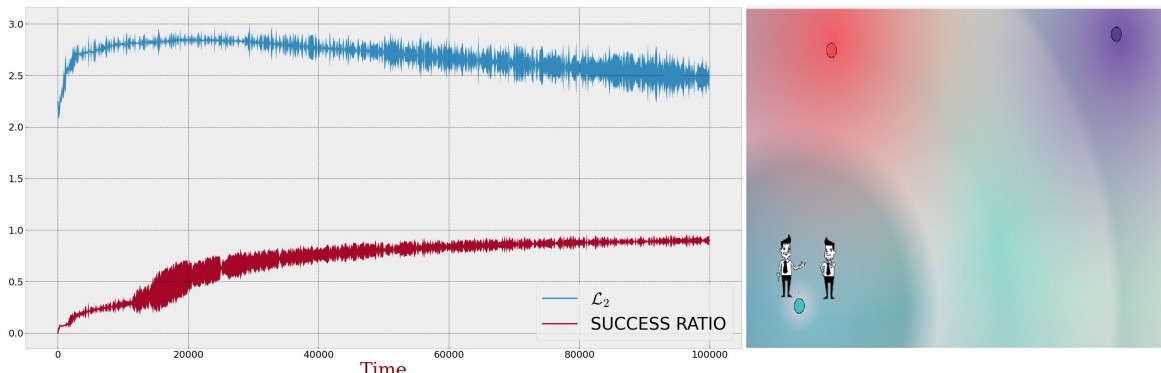

**Figure 13:** Success ratio for the setting with $N = 3$ vertices and 2 homogeneous agents and the same concept space as in the previous setting

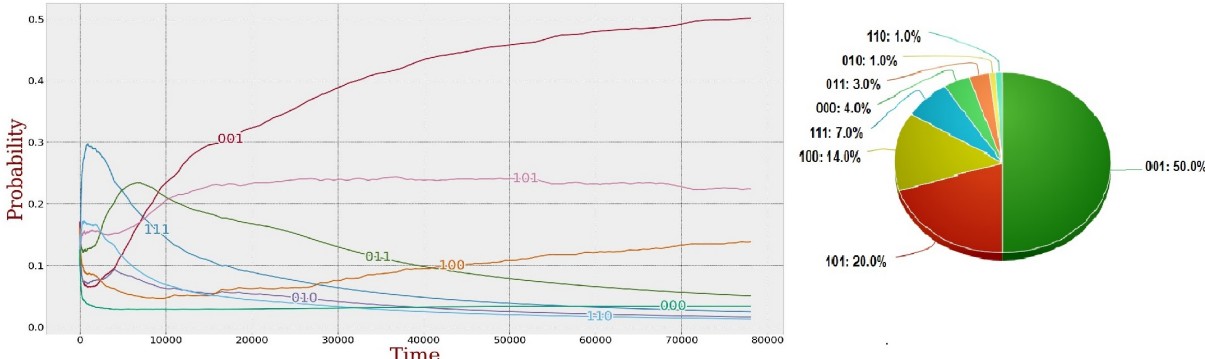

**Figure 14:** Emergence of concept selection pattern for the setting with $N = 3$ vertices and 2 homogeneous agents with the same concept space as in the previous setting. The plot illustrates the frequency ratio of the chosen concept selection during the dialogue utterances. The continuity in the trajectories indicates that the distribution over the concept selection space remains consistent across the agents.

the impact of the principle of least effort on the Zipfian characteristic of emerged languages. Specifically, we looked at scenarios where the weight (represented by $\kappa_2$) associated with the loss function that quantifies the principle of least effort varied (Figure 9). When $\kappa_2$ had higher values (1.1 in one case), the discrepancy between the observed word usage frequency and the expected Zipf distribution is minimal. This indicates that the sensitivity of the Zipfian characteristic to the weight of the principle of least effort. In other words, higher weights lead to word frequencies that more closely follow an inverse relationship to their ranks, as predicted by Zipf's law.

## 6.6 Language emergence at scale

The agent population is upscaled to observe the emergent behaviour among large populations. We consider complex settings with the number of different agent pairs equal to $12, 30$ and $42$. The emergence of a shared language among a larger population is cumbersome which requires a large number of iterations. The underlying policy gradient algorithm develops coherence by reinforcing successful interactions. However, in the case of more agent pairs the probability of propagation of mappings involving successful interaction among the population is minimal. This behaviour is illustrated in Figures 17. Hence at scale, language emergence among larger populations proves challenging, necessitating numerous iterations to propagate successful interactions and generate shared understanding.

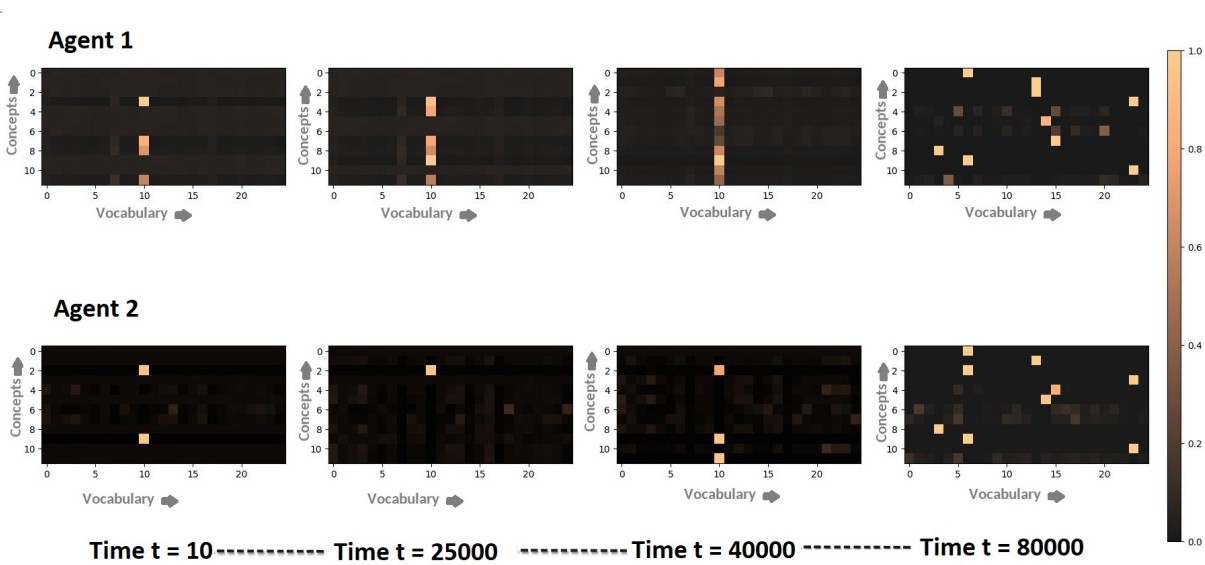

**Figure 15:** Success ratio for the setting with $N = 3$ vertices and 2 homogeneous agents and the same concept space as before

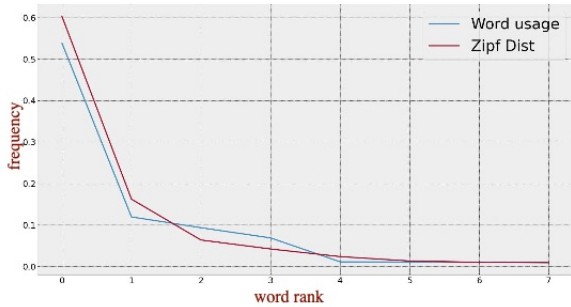

**Figure 16:** The usage of a word reduces in accordance with their rank supporting the Zipf law.

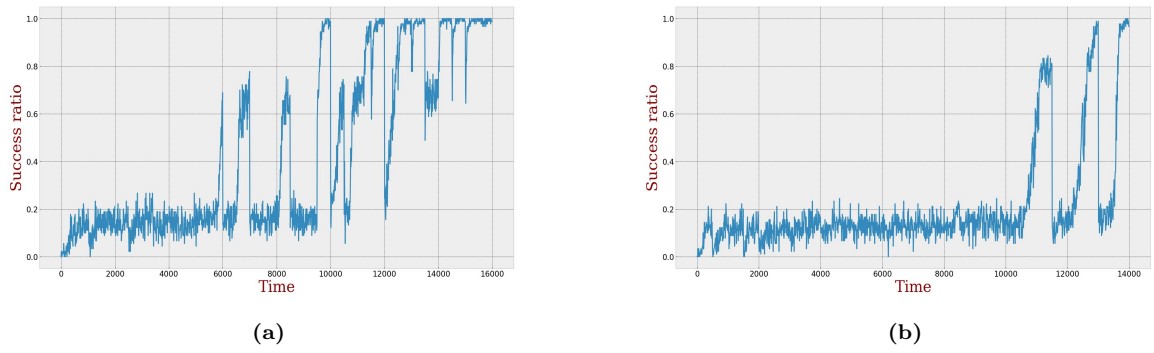

**Figure 17:** (a) Plot of success ratio with 30 pairs of agents and 5 vertices. (b) Plot of success ratio with 42 pairs of agents and 5 vertices

# 7  Conclusion

In this paper, we develop a computational language game framework to model the factors influencing language dynamics involving a finite number of homogeneous deep neural agents in a guessing game setting. We factored silence as a symbol for optimal communication, guided feedback scenario to consider poverty of stimulus. We observe the successful emergence of grounded vocabulary and compositional language structure among agents. Our experimentation involved varying the population, vocabulary and concepts sizes to systematically observe these emergent linguistic patterns. Notably, our findings align with natural phenomena, demonstrating properties such as the principle of least effort, Zipf's law, and the synchronization of inverse mappings.

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
