# OpenReview forum: "Emergence of Grounded, Optimally Compositional Spatial Language among Homogeneous Agents"
_TMLR — Rejected by TMLR_

### Review · Reviewer_2sqP · 2024-01-22

**Summary Of Contributions:**

This paper presents an emergent communication study involving (mostly) two neural network agents playing a situated "guessing game" where the task is to communicate the location and color of different vertices of a graph embedded in a 2D plane. The authors design neural network agents to serve as speakers and listeners for this game, and carefully control the concept space and introduce several auxiliary losses designed to get the agents to communicate in grounded and compositional manners.

Indeed, on their guessing game, the authors find that the emergent languages have several desirable properties that are also in human languages, e.g.:

1. As a result of a length penalty, the messages obey Zipf's law;
2. There is some convergence on a shared set of signal/meaning relations (groundedness);
3. As well as some notion of compositionality.

**Audience:**

Yes

**Claims And Evidence:**

No

**Requested Changes:**

Overall, I don't think the paper is quite ready for publication yet. I think to secure my recommendation, the following things would need to happen:

1. Better compare/contrast to the existing EC literature, both in the Introduction (/related work), during the explanation of the environment (and why these differences exist), as well explaining how the conclusions differ.
2. The paper should more carefully control and understand the various design decisions made and how they do/don't effect the emergent languages, rather than just running a single monolithic experiment. This will help researchers glean more generalizable conclusions from the results.

**Strengths And Weaknesses:**

# Strengths

- The paper suggests several interesting and well-motivated ways to steer the communication of agents more towards human-like languages. My only issue is that these various ideas aren't explored in a well-controlled manner—rather, a "kitchen sink" approach is taken where the results of one study, with all of these compounding changes, is presented (see weaknesses).
- The authors introduce a novel  signaling game environment with more complexity than existing Lewis signaling games, that may be of interest to the rest of the community.
- The authors conduct an extremely thorough analysis of the resulting languages produced by the agents, with several interesting visualizations and conclusions (e.g. Figures 8, 9).

# Weaknesses

My biggest issue with the paper is that it is currently presented as one large, monolithic EC study, with considerable complexity and many design decisions that differ from existing work. As a result, it's sometimes unclear what one should get from the conclusions of the paper: since only one setting has been explored with several independent variables changed, it's not clear whether the findings (e.g. "Obeys Zipf Law, a few concepts are dormant,") generalize in a way that is useful to other language evolution/ML/RL researchers. I think this is the result of a few converging problems which I'll explain below.

## Lack of carefully controlled experimental conditions and ablations

Again, because so many independent  variables have been tweaked and modified, it's not clear which component(s) of the experimental design are responsible for the different phenomena observed, and thus it's not clear which experimental changes have the possibility of generalizing and being of interest to people working on environments besides this particular one. For example, the authors claim the agents language "Obeys Zipf law" due to the loss function operationalizing principle of least effort—is this true? To be confident, we would want to see that the emergent languages are not zipfian without the loss, but zipfian with.

There are *some* additional experiments described in Section 6 (e.g. point 9 on compositionality, point 11), but overall the explanation of the experiments is rather brief, and I think such experiment(s) could be expanded, and the same rigor applied to the various other choices in the environment, e.g. loss function, training regime, etc. In particular, the loss function is composed of 3 different parts—are the various losses indeed doing what they expect to the resulting language, if any of them are turned off?

## Unclear contribution over existing work

To strengthen the paper, the paper should also explain better how its conclusions are different/novel from the existing EC literature, especially given that the study has so many design decisions that veer off of the standard Lewis signaling game track. Currently, the paper doesn't adequately compare and contrast to existing work.
There are a large number of citations in the second paragraph of the introduction, but the previous papers are simply listed, rather than compared with the present study. For example, the last paragraph of the introduction claims:

- "This is the first attempt of its kind to study language emergence in guessing games." This seems false, unless the authors are using a highly specific definition of "guessing game" that excludes the large body of work on Lewis signaling games. And even if the authors claim that this is the first work on this kind of guessing game in particular, more time needs to be spent explaining why this is important, and why this allows us to answer questions that the existing literature in EC has not yet addressed.
- "...as well as homogeneous multi-agent settings." This also seems false, especially because the vast majority of experiments in the paper only uses 2 agents (although the paper seems to suggest that it is novel in part because it looks at N >= 3 agents, this is stated in the first paragraph of page 10). Most of the work cited in the paragraph above also studies two-player communication games, and many even study N+ player for N > 2.
-  "We also introduce efficient communication through the principle of least effort.". Again, this is also not clear, [Chaabouni et al. (2019)](https://proceedings.neurips.cc/paper/2019/file/31ca0ca71184bbdb3de7b20a51e88e90-Paper.pdf) explicitly looks at to what extent EC languages are Zipfian, and whether length penalties steer the languages more towards Zipf's law. This is not the first EC study to analyze these properties, and the paper needs to more clearly compare and contrast with their findings.

## Complicated environment, unanswered questions

 Overall, I found the presentation of the game fairly complicated. Here are some questions I would like clarified about the environment after reading:

1. Agents only have a partial observation of the environment, as stated at bottom of page 2. What is that partial observation function $f_i$?
2. Is the speaker objective differentiable or not? End of paragraph 1 in sec 4.1 is a bit confusing to me, where the network utters a message via standard non-differentiable categorical sampling, but then gumbel softmax is explained as "enabling differentiability of concept-selection to utterance channel."

Again, because the game is complicated, I feel this means the burden is on the authors to explain and justify why the complexity is necessary to answer the questions the authors seek to answer. (Again, better contextualizing the paper with existing work, and explaining why and how this paper differs, is key here.)

## Minor

- There are a lot of references to cogsci/linguistics/neurosci literature which I'm not sure are entirely warranted. e.g. sec 5.3 mirror neurons are cited, but what is really happening is simply a loss to encourage the representations from the utterance + listening network to be similar; I'm not sure mirror neurons is the right comparison here since those are highly specialized, and in this case the entire representations are just pushed together. Moreover, variatns of this idea have also been implemented before, e.g. [Choi et al. (Compositional Obverter Communication)](https://arxiv.org/abs/1804.02341).
- end of page 7: "by congruence, we mean in the Bayesian probabilistic sense". I'm not sure what this means.
- The part that involves multiple agents is extremely brief, just briefly touched upon in point 11 of section 6. This is too brief and many details surrounding the multi-agent implementation(s) are missing. I suggest either authors expand this section substantially, or cut it out and also cut out verbiage in the rest of the paper that seems to suggest that the paper primarily focuses on the multi (N >= 3) setting, since most of the paper focuses on dyadic interactions (which is very similar to most of the existing EC literature)

---

> ### Author Response · Authors · 2024-03-12
> **response to Reviewer 2sqP**
>
> We thank the reviewer for the valuable comments.
>
> ## Lack of Carefully controlled experimental...
> All the controlled experiments and ablation studies are addressed in the Section $6$ of revised submission.
>
> ## Unclear contribution over existing work
>
> *"This is the first attempt of its kind ..." This seems false ... definition of "guessing game" that excludes the .....* --- The more comprehensive explanation and reasons pertaining to considering guessing game and its complexity are added in Section $2$ and $3$. The guessing games are inherently complex due to the combinatorial explosion arising due to the meaning uncertainty during failed conversations. This makes construction of a shared vocabulary by the alignment of the meanings through communication the more difficult.
>
> *"...as well as homogeneous ... settings." This also seems false, especially ... vast .... , this is stated in the first paragraph of page 10). Most of the work cited in the ..... --- In our updated experiments we have considered a more complex setting involving $12$ agent pairs ($4$ agents).
>
> *"We also introduce efficient ... principle of least effort.". Again, this ... not clear.... extent EC languages are Zipfian, .....* --- (Chaabouni et al. (2019)) investigates the effect on the message length for the given input, where input are sampled following a power-law distribution, where it is observed that initially the larger length messages are used by less frequent input with the inputs and language being non-compositional. In our setting, the concept selection module decides upon the required/essential concepts which defines the topic vertex. The concept selection module optimises the conceptualisation effort/cognitive economy by performing description length minimisation (with entropy regularization). The economy of thought gradually transpires into compositionality in the emergent language since language encapsulates thought. Hence one could observe the emergence of efficient communication strategies, where shorter phrases are preferred over longer ones. Further it is observed that varying the attached weight to the principle of least effort loss, the emerged word usage closely follows Zipf's law which can observed in Figure $9.8$.
>
> ## Complicated environment, unanswered questions
>
> *Agents only have a partial observation of the .... What is partial observation function?*---
> The partial observation function $f_i$ gives the partial observation vector $o_i$ depending the current state of the environment. In our experiments, the considered observation vector $o_t$ is defined in the Section $4.1$
>
> *Is the speaker objective differentiable or not? End of paragraph 1..."* ---
> The speaker objective is designed to be differentiable, and we utilize the Gumbel Softmax technique to ensure differentiability in the channel between the concept-selection and utterance modules of the agent. This choice is justified by the fact that this channel operates within each agent, rather than between agents. The outcome of the interaction must update all parameters within the speaker agent, fostering inter-dependency between its components. However, the channel between the speaker and the listener is discrete and thus non-differentiable. This characteristic ensures decentralization in our setting, where agents are independently calibrated, preventing any correlation from being established between them during learning, akin to avoiding a puppetry show. Shared communication is driven solely by the shared reward.
>
> ## Minor
> *There are a lot of references to cogsci literature which ... warranted. e.g. sec 5.3 mirror neurons are cited, .....; I'm not sure mirror neurons is the right comparison ... case the entire representations are just pushed together.* ---
> We apply mirror networks to enable continuity  in the evolved language during the role switching of the agents (We refer to this as homogenity/interchangeability). This ensures that agents learn the reverse bidirectional mapping along with their active role mappings. In [Choi et al. (Compositional Obverter Communication)](https://arxiv.org/abs/1804.02341), we observe a psuedo-homogenity where each agent possesses only one network which is updated while listening to align itself with the speaker. While speaking, the speaker agent goes over all the symbols which best matches the input by running it through the network. Also, during interactions, only listener's parameters are updated and speaker's parameters are fixed based on the assumption that speaker is correct every time.
>
> *end of page 7: "by congruence, we mean ....* --- posterior distribution of the parameters conditioned on the message or concept (Updated in the manuscript)
>
> *The part that involves multiple agents is extremely ....* --- In the update revision, our base experiments consist of $12$ agent pairs ($N=4$ agents). This is more than the number of agents considered in the existing literature which mostly involves 2 agents.

---

> > ### Author Response · Authors · 2024-06-12
> > **Response 2**
> >
> > We kindly ask the reviewer to review the updated manuscript. We have made significant revisions based on the major comments received during the initial review. The crux of our study lies in exploring the complexity stemming from dynamic interactions, coupled with the guessing game element and decentralized learning. Despite this complexity, our study demonstrates the emergence of a language with notable characteristics such as groundedness, shared understanding, coherence, compositionality, word order, and a Zipfian distribution.

---

> > > ### Comment · Reviewer_2sqP · 2024-06-24
> > > **Update**
> > >
> > > Hello, apologies for the late reply.
> > >
> > > Thank you for your significant effort in updating the manuscript, especially adding section 6 which adds significant ablations over the original paper. This is a marked improvement and I appreciate how much work was put into the revision.
> > >
> > > Overall I still share the sentiment that it is somewhat unclear what the aims of the present study are, and how the study aims to answer questions that existing work in EC has not. The expanded section 6 has the impression of "lots of varied experiments and ablations, but somewhat unclear what the key takeaways are for people studying language emergence outside of this particular environment."
> > >
> > > The added section 1.1. contributions says:
> > >
> > > > ...we develop a game setup allowing significant
> > > complexity compared to the existing Lewis signalling games settings in terms of combinatorial possibility of
> > > mapping the words to a concept by listener for the novel words uttered by the speaker. Additionally, we explored the notion of interchangeability property in the language which enables agents to simultaneously synchronise the bidirectional mappings along with their active role (either listening or speaking) in the game which ensures continuity and internal consistency. In this paper, we also introduce efficient communication through the principle of least effort, where the agents are encouraged to convey information in a way that minimizes complexity or cognitive effort...
> > >
> > > Again, I think there are simply too many experimental variations at play here, and actually many of these variations could be quite carefully studied and perhaps even end up being papers in their own right. Even the first innovation here:
> > >
> > > > We develop a game setup allowing significant complexity compared to the existing Lewis signalling games settings in terms of combinatorial possibility of mapping the words to a concept by listener for the novel words uttered by the speaker.
> > >
> > > Could be carefully investigated and controlled. Why do existing Lewis signaling games not provide sufficient complexity? Why is the present EC environment the right way to ramp up this complexity? Why and how does this complexity mirror human language evolution? How does the language from a speaker/listener smoothly change as we vary the combinatorial complexity of the underlying game?
> > >
> > > It's possible to answer plenty of questions in this space while keeping all other factors of variation fixed. However, there are many more contributions here ("interchangability", "principle of least effort", ...) etc. By the time the entire study has been built up it's unclear which factors led to which observed phenomena. The ablation studies in Section 6 help, but overall none of the proposed changes here are fully isolated. The authors themselves state in Section 6 that the overall takeaway is that:
> > >
> > > > Thus in the intricate tapestry of our decentralized guessing game involving multiple agents, diverse phenomena weave together,
> > > depicting the emergence of a shared, grounded, structured and efficient communication system.
> > >
> > > Again this idea of "diversity" and "intricacy" is a double-edged sword: it seems like many interesting phenomena were observed here, but the game is so complex that it's unclear whether such results generalize.
> > >
> > > Many of the motivations for creating an "intricate" guessing game are ostensibly rooted in the linguistics literature, but each of the particular innovations are by themselves somewhat sketchy or could receive pushback from people (e.g. see other reviewer discussion on mirror neurons). At the end, it's somewhat unclear whether the insights in this paper are generalizable to either cognitive scientists or people studying MARL in different environments, because (1) the environment is far removed from existing work in EC or realistic MARL tasks, and (2) how these changes actually operationalize human language use is somewhat up for debate, especially when multiple changes are combined together into a large environment.
> > >
> > > I think rather than "presenting a monolithic study", then attempting to take away things/measure ablations in Section 6, the authors could consider a more bottom-up approach, where the study is constructed by first thinking of more concrete hypotheses of interest in either multi-agent RL or language evolution, then building a single study to test that hypothesis.
> > >
> > > ---
> > >
> > > Overall I would echo some comments by the other reviewers:
> > >
> > > Reviewer Xenq:
> > >
> > > > To clarify: I did not expect a direct experimental comparison of other models in the literature, but a motivation of all parts of the current model by referring to standard approaches and techniques and pointing out why they are not sufficient and how the current paper improves upon them.
> > >
> > > EwVC:
> > >
> > > > ...there are a large and varied set of choices made in implementation, and it would be helpful to know which are important and which are not. This would, for example, help facilitate choice of control conditions to test hypotheses.

---

### Review · Reviewer_EwVc · 2024-01-23

**Summary Of Contributions:**

The topic and approach of the paper are interesting. In particular, what are the minimal set of principles in order for human-like language to arise, is a fascinating question. The approach adopted in the paper is a bit different, however, focusing on some fairly specific assumptions about how the problem could be solved.

**Audience:**

No

**Claims And Evidence:**

No

**Requested Changes:**

Based on the above, I would ask for a comprehensive revision of the paper beginning with a clear statement of the hypothesis that was being tested, and a clearer positioning of the paper vis a vis the broader related literatures. (I'm sorry that there are so many!) With those two matters addressed, there would remain the question of how the simulations supported the account, which would also need to be evaluated.

**Strengths And Weaknesses:**

One weakness of the paper is that it was unclear what the paper intended to show. The abstract includes claims such as language is an "efficient, unambiguous, adaptive, and coherent apparatus to convey one’s goal to others" and "can emerge in a finite population through incremental
learning via trial and error at the individual (micro) level, with nearly consistent individual
learning faculty and experience across the population" which might be better stated as hypotheses?
The last paragraph of section 1 states "we develop a computational language game framework to model the factors influencing language dynamics involving a finite number of homogeneous deep neural agents with sensory-motor abilities who wish to convey their goals to other agents effectively through communication using deep reinforcement learning in a guessing game setting" This does not say why we are doing this, which is important because we hope to learn something from this study. (That is the why.) There are also the claims that this is the first attempt to study language in guessing game settings, which is an extremely bold claim that is very hard to believe. There are substantial literatures on guessing games and language learning! Also, there is the claim that the introduction of efficient communication through the principle of least effort is first of its kind is not believable (or at least would need to be clarified). There is quite a literature on efficiency in communication. Aside from these arguments, I couldn't find a clear statement of what we hoped to learn from the study. This is important because there are a large and varied set of choices made in implementation, and it would be helpful to know which are important and which are not. This would, for example, help facilitate choice of control conditions to test hypotheses.

Another big weakness is that although human language is discussed a great deal, the actual connections to humans or human language are minimal. Perhaps the biggest point of departure is that language is unambiguous. This is famously not true of human language. The entire field of linguistic pragmatics is dedicated to studying this, though only a subset of the true ambiguities of language. A second is the invoking of mirror neurons. This topic has been the subject of rich debate, with quite strong arguments against both theoretically and practically. I have included a few pointers to that debate below, which is important to contextualize the work. The mirror loss itself seems to have little to do with mirror neurons though, in fact it is more closely related to other claims in the literature such as RSA and rational imitation (again some pointers are provided below). Similarly, poverty of the stimulus has been discussed for a large number of years with a number of approaches previously proposed. Classics include the subset principle by Bob Berwick, which was updated in Josh Tenenbaum's dissertation and applied to language by Andy Perfors (among others). Similarly, there have been a number of studies of guessing games, mostly in linguistics, but also in cognitive science.


Detailed questions
- "glossogenetics" It would be good to define the term when using it the first time.
- "primal human communication mechanisms," What does the "primal" part mean?
- "mirror networks" The reference to mirror networks was hard to understand. Reading through the remainder of the paper, I might understand the idea, but it would be helpful to define the term.
- "through consistent learning" What does the "consistent" part mean?
- "This is the first attempt of its kind to study language emergence in both guessing
games as well as homogeneous multi-agent settings." This seems like a bold claim. I would really like to see a more focused introduction making the case for what is the related work in language games and multi-agent settings language learning settings.
- "In this paper, we also introduce efficient communication through the principle of least effort. This also is the first of its kind." This claim is too strong. As the authors cite, Zipf published his dissertation with this idea almost a century ago. It is safe to say it is one of the most researched topics in language since then. Most of the attention has focused on the distribution itself. But there is also quite a substantial literature on least effort in language. I have included a citation below, but it is far from the only one. I would strongly implore the authors to review the literature.
- "In case of failed communication, the speaker discloses the target node to the listener." This is a big difference from the human communication that was featured in the introduction.
- "Mirror neurons" are a pretty controversial topic. Also, they have little resemblance to the computational problem assumed. It would be helpful to discuss the literatures in which these topics have been discussed. There has been a broad debate about simulation theory vs theory theory in psychology that has touched significantly on the interpretation of mirror neurons. There is a literature on rational imitation, which is related to inverse reinforcement learning, which argues against mirror neurons as providing significant explanatory power. There is a broad literature on communication, specifically RSA (rational speech act) models and their ilk that are related as well. I think you will find that the optimization problem posed under that section has interesting connections to models of cooperative communication and common ground.
- "The hyper-parameters (learning rate, batch size, and regularization strength) are fine-tuned through iterative experimentation" What should readers think of this?
- "reward values for various scenarios are obtained through an exhaustive, yet rational search" What is rational about the search?
- "The concept space C consists of 4 sectors, 3 segments and 4 colors." There are a discrete and finite set of concepts?


Piantadosi, S. T., Tily, H., & Gibson, E. (2011). Word lengths are optimized for efficient communication. Proceedings of the National Academy of Sciences, 108(9), 3526-3529.

---

> ### Author Response · Authors · 2024-03-06
> **response to Reviewer EwVc**
>
> We thank the reviewer for the valuable comments.
> *The last paragraph of section 1 states "we develop a computational language game framework to model the factors ....." This does not say why we are doing this, which is important because we hope to learn something from this study. (That is the why.)* --- We have defined the guessing game and relevant details at the end of Section $3$ of the updated manuscript.
>
> *"glossogenetics"* --- The term Glossogenetics refers to the study of the origins and development of language.
>
> *"Primal human communication mechanisms"* --- Refers to a emergent communication from a population with no existing predefined language.
>
> *"Through consistent learning"* --- It is used in the sense that agents will keep updating their parameters to update their learning from the conversations.
>
> *"mirror networks" The reference to mirror networks was hard ..... but it would be helpful to define the term.* -- We have defined mirror networks in Section $5.3$ of the updated manuscript.
>
> *"This is the first attempt of its kind to study language emergence in both guessing games as well as homogeneous multi-agent settings."* --- This combined dynamics $i.e.,$ (homogeneous agents and guessing game) together is studied for the first time in the field of emergent communication. We have updated the manuscript with more details (Last paragraph of introduction section)
>
> *"In this paper, we also introduce efficient communication through the principle of least effort. This also is the first of its kind." This claim is too strong. As the authors cite, Zipf published ....* --- We investigate the emergence of efficient communication through principle of least effort in the emergent communication setup by introducing the principle of least effort during the dialogues between the agents. The relationship between them as rightly pointed out by the reviewer is already present. However, we investigate this relationship in the emergent communication setting involving multiple agents through deep reinforcement learning to corroborate the veracity of the claim by performing ablation studies. Note that this is one of the components of dynamics involving language emergence in our setting.
>
> *"In case of failed communication, the speaker discloses the target node to the listener." This is a big difference from the human communication that was featured in the introduction.* --- In any human communication the verbal communication is supported by the physical or tangible communication which we also incorporates in our setting. We do not consider telepathy between speaker and listener rather a tangible feedback from speaker to listener.
>
>
> *"Mirror neurons" are a pretty controversial topic. Also, they have little resemblance to the computational problem assumed. It would be helpful to discuss the literature in which these topics have been discussed. There has been a broad debate about simulation theory vs theory theory in psychology that has touched significantly on the interpretation of mirror neurons. There is a literature on rational imitation, which is related to inverse reinforcement learning, which argues against mirror neurons as providing significant explanatory power. There is a broad literature on communication, specifically RSA (rational speech act) models and their ilk that are related as well .....*  ---
> Mirror networks present in the agent architecture enables bidirectional mappings which are used to support the interchangeability property of the language where  agents can  switch the roles between speaker and listener. This enables them to utilize the information learnt during comprehension to reproduce the language and vice versa. To ensure that the agent learns the reverse mapping between concepts and words, we employ mirror networks which are synchronized with the active networks via reducing the KL divergence between the posterior probabilities. We refer them as mirror networks to reflect the fact that they are conditionally opposite.
>
> *"reward values for various scenarios are obtained through an exhaustive, yet rational search" What is rational about the search?* --- In reinforcement learning settings, the optimal reward values are typically determined through trial and error methods. In our experiments, we also investigate various reward functions to identify those that best enhance success in the language game. This approach is a common procedure for determining the appropriate parameter values for a specific setting. (Satwik Kottur, José Moura, Stefan Lee, and Dhruv Batra. Natural language does not emerge ‘naturally’in
> multi-agent dialog. Conference on Empirical Methods in Natural Language
> Processing 2017)
>
> *The concept space C consists of 4 sectors, 3 segments and 4 colors." There are a discrete and finite set of concepts?* - Yes these concepts are discrete and finite.

---

> > ### Comment · Reviewer_EwVc · 2024-05-22
> > **Thank you for the response**
> >
> > Thank you to the authors for a detailed response. The responses locally address the questions without considering the broader implications for the motivation and structure of the investigation. After re-reading all of the reviews and the responses, my impression of the paper is unchanged.

---

> > > ### Author Response · Authors · 2024-05-25
> > > **Response -2**
> > >
> > > I appreciate the reviewer's feedback
> > >
> > > I kindly ask the reviewer to review the updated manuscript. We have made significant revisions based on the major comments received during the initial review. The crux of our study lies in exploring the complexity stemming from dynamic interactions, coupled with the guessing game element and decentralized learning. Despite this complexity, our study demonstrates the emergence of a language with notable characteristics such as groundedness, shared understanding, coherence, compositionality, word order, and a Zipfian distribution.
> > >
> > > While drawing direct comparisons between our findings and real-world human language scenarios presents challenges and may be somewhat unfair given the inherent complexity of human communication, it remains essential to evaluate our results within the context of existing artificial language emergence studies. We extensively review such frameworks in our introduction.

---

> > > > ### Comment · Reviewer_EwVc · 2024-06-14
> > > > **Response**
> > > >
> > > > I did review updated the manuscript.

---

### Review · Reviewer_Xenq · 2024-04-26

**Summary Of Contributions:**

**Summary:**

The paper conducts a case study of training a population of 4 LSTM-based agents in a simple environment that consists of 5 nodes that each have a (coarsely discretized) 2D position and one out of four colors. Later this is extended (with mixed success) to max. 7 agents and up to 50 vertices. Each agent is located on a node, and pairs of agents are randomly grouped into speakers and listeners (each agent has both roles throughout the game). The listener picks one of the nodes (except its own location) and produces an “utterance” of a sequence of max. 3 tokens to describe the node (relative from the speaker’s position) to the other agent, the listener. Both agents are rewarded if the listener successfully identifies the correct vertex; there is a lower reward if some properties are correctly identified; and an even lower (negative) reward if the listener fails. The aim is to train all agents (in a decentralized RL fashion) to successfully communicate with each other all the time, which requires a shared vocabulary of utterances that is “spoken” and “understood” by all agents. The paper demonstrates a setting where this is possible by adding many more components to the agent architecture and training setup (multiple regularizers, training on different time-scales, …). This setting is studied in detail and various aspects of this “emergent language” are related to important aspects of human language.

**Main contributions:**

* Design of a simple environment to provide “grounding” for an emergent language in the agent population, and design of a reward function that is maximized by a shared “grounded” vocabulary among agents.
* Design of LSTM-based agents that can be both speakers and listeners. This involves being able to process an observation from the environment into an utterance (two LSTMs are involved here, one to pick the aspects of the node to communicate, and one to produce the actual utterance). Another LSTM implements the listener, that takes in another agent’s utterance and outputs position (distance and angle) and color of the best-guess target node. In principle speakers and listeners could be completely separate populations, though there is a loss regularizer that facilitates representational similarity between speaking and listening LSTMs within each agent - if speakers and listeners were separate populations, this loss would break the decentralized learning assumption and could not be used.
* Development of a complex training protocol (including multiple losses and auxiliary reward signals, various hyper-parameters, different timescales of training the two speaking LSTMs, and stochastically having the speaker providing the ground-truth to the listener in case of a failed communication) that leads to the desired result in this particular case.

**Audience:**

No

**Broader Impact Concerns:**

No broader impact concerns.

**Claims And Evidence:**

No

**Requested Changes:**

**Improvements:**

The weaknesses pointed out earlier need to be addressed:

* Formulate a set of falsifiable claims and hypotheses and empirically verify them.
* Critically analyze the findings and rule out alternative hypotheses that would explain them.
* Improve the manuscript for clarity (see my comments below)
* Fix the mathematical notation and formalization in the paper (see my comments below)
* Formalize abstract concepts (from linguistics or otherwise), make them precise, and connect them to concrete math. expression, well-formed hypotheses, and experimental observations (and show or hypothesize how alternative observations would not correspond to, or violate, the corresponding concept).
* Tone down the connections to human language and linguistics, or improve these connections by aligning more closely with the literature, models, and practices in modern comput. linguistics and neuroscience.
* For each auxiliary loss- and reward-function, and for the various parts of the agent architecture: perform ablations to show the qualitative and quantitative effect that leaving out this part of the architecture has and that the corresponding part actually plays the abstract role that it is connected / motivated by.
* Give full details (in the appendix) of all hyper-parameters and other settings required to reproduce the experiments.
* Answer the following questions in the manuscript:
  * What is the problem this paper is attempting to solve? Which open question(s) is it addressing?
  * What is the state-of-the-art w.r.t. the main problem/question? What is lacking in previous attempts (published environments, agent-architectures, and training procedures), and how does the current paper improve upon that?
  * What concrete generalizable findings can be taken away from this study?

* [I consider this an optional, though significant, improvement] The language constructed is finite (which means a look-up table, with no intelligence or cognitive faculties whatsoever easily suffices to “speak and understand” the language in principle). Ideally, the language game would use an infinite regular language (or even better context-free, but that poses a substantial learning problem); in an infinite language (where an infinite number of finite words can be produced from a finite alphabet), “deep” understanding of the meaning of words and utterances can very easily and convincingly be tested by presenting unseen words from the language to agents (here is where a look-up table will fail, but an agent that has acquired language understanding will succeed).



**Minor comments** (in no particular order; I have also pointed out some questions I had when reading that are answered later in the manuscript - hopefully these questions still help to improve clarity and potentially reorganize paragraphs and sections, but they do not need to be answered in the rebuttal):

* What makes the language game in Fig. 1 “special” - why could the same conclusions not be drawn from any cooperative, Markovian multi-agent interaction with partial observabitliy?

* Please consider the difference between \citep and \citet (many references and groups of references should be put in parentheses with \citep for improved readability).

* Problem formulation: what precisely are “coherent symbolic structures”?

* Problem formulation: “presenting the utterance using an appropriate conceptualization and vocal language on a noise-free, face-to-face, discrete channel where everything said is heard” - this is very ambiguous (even together with the figure) and needs to be made more concrete. Also, why is face-to-face relevant if the channel is noise free - if not, remove this qualifier.

* I cannot connect the problem formulation and Fig. 1 to the generic description of cooperative RL in a Markov environment (last paragraph on page 3). In Fig. 1, what is the state of the world $s^{(t)}$. Also, what exactly are the action- and observation-spaces (which are different for speaker and listener?). These questions are mostly answered later in 4.1 Policy Architecture, but should be stated earlier when introducing the Markovian environment.

* “The segments and sectors provide a conceptualization of space that is grounded in the sensory and physical interactions of the agents with the world and one can relate it to the concepts of cardinal directions in the human discourse.” - that is a big stretch. The agent perceives both as simple integer numbers and one could randomly permute these numbers to destroy any topology while artificial agents would still be able to play the language game perfectly well.

* Formalism has many mistakes and is inconsistent between text and Fig. 3 (and even inconsistent within Fig 3, such as $h_0^s$) and is missing important details - this must be cleaned up. Below is what I spotted mainly in Sec. 4.1 and Fig. 3:
  * $x^{(1), \ldots, (M)}$ instead of $x^{(1), \ldots, (N)}$, and same for $q$.
  * $z^{(i)} \in {1, \ldots, M}$ instead of $N$.
  * Does $z_t$, the location of each agent change in each time-step. If yes, how? Otherwise remove the $t$ subscript from $z$ in the state vector.
  * $q^{(i)} \in \mathbb{R}$ - why is q written as a bold vector when it is a scalar in $\mathbb{R}$?
  * $u_t^{(i)} \in ~?$
  * Speaker $o_t^{(i)}$: why is  $z_t^{(i)}$ time-dependent, but $d$ and $w$ are not?
  * Fig. 3 How is $h_0^s$ defined?
  * Fig. 3 How is $h_0^l$ defined and what is the observation vector $o_t^{(i)}$ for the listener?
  * $m_t \in ~?$ - is the utterance a vector in $\mathbb{R}^3$?
  * $b_t \in ~?$
* Is the concept selection bit-vector really a vector of **binary** bits? This is inconsistent with $\mathcal{L}_2$ in Sec. 5.5 where the description-length loss has an L2-Norm penalty over $b$. Please explain or choose a better name for $b$.
* Why is there Gaussian noise on $d$ and $w$ (the distance and angle of vertices)? Is it needed to introduce variety (entropy) into the utterance module by making the concept selection bits more random? If yes, why not inject the noise directly into the utterance module via $c_i$? What happens to the results when the noise is removed?
* Why is the concept selection module needed? Could it not be folded into the utterance module by passing the initial hidden vector directly into the utterance module along with $c_t$?

* 5.1: It looks like guidance is weakening the notion of decentralized learning. If $\lambda$ were always set to 1, the setup would reduce to a **centralized** learning problem. The question then is, how important is the guidance; if it is crucially necessary, then the setting could also be considered a noisy centralized learning problem (missing ablation).

* 5.2: Mirror loss needs an ablation. What happens empirically and theoretically without the mirror loss. What is the falsifiable hypothesis?

* 5.2: Sentence length is only one factor related to the effort of producing and understanding language. Relating it as the sole factor to a principle of least effort is impoverished (I personally would argue that maximizing encoded information with sufficient redundancy to be transmitted via a noisy natural communication channel are the far more important considerations than sentence length).

* 5.5: There is a mistake in the signs of the loss terms. Assuming that $J$ is maximized, then  $\mathcal{L}_1$ and  $\mathcal{L}_2$ look correct (though  $\mathcal{L}_1$ is not a loss, but a utility term) but  $\mathcal{L}_3$ needs a negative sign (because it is supposed to be minimized).

* Sec 6: “The feasible reward values for various scenarios are obtained through an exhaustive, yet rational search.” - what does this mean exactly?

* Page 13 (bottom): broken Figure reference.

* Fig. 8, a,b: x-axis is not time but number of dialogs.

* Ideally show that results are qualitatively the same across different training runs; since a lot of the agent architecture and training setup is geared towards leading to robust emergence of shared, grounded vocabulary, this must be empirically demonstrated and quantified. Merely showing results from a single successful run and studying that run’s properties extensively is not enough to support claims of generality.

* 6.3.2 Emergence of a word for silence: the text suggests a surprising emergence of a shared word for silence, while this is actually forced via a significant auxiliary loss (Eq. 7) that rewards individual agents to reuse the same word for silence (across concepts) and agents are forced to coordinate their vocabularies globally (and use short sentences).

* 6.5.1 This is an observation for a single training run - what generalizable finding can be taken away?


* [after reading up to Sec. 4, I had the following questions; they are answered later in the manuscript but consider pulling these answers further up] Do agents actually move around in the (graph) world? When they are on one node, do they “see” many sectors and segments (some with colored dots) that can only be seen when on this node in the graph, or is each colored dot (in Fig 2d) another node in the graph? What are the agents’ actions - are there actions for moving in the (graph) world and is the state of the graph world the location of each agent? Or are actions utterances (speaker) and pointing to targets (listener)?

* Problem formulation [this is answered later in the manuscript]: “Our agents are homogeneous in nature, where they can perform both comprehension and reproduction of language.” What precisely does this mean? Is this a statement about learnability of language by the agents in principle?

* Problem formulation [this becomes clear later in the manuscript]: What is a “semiotic cycle of dialogue”? And what distinguishes it from a non-semiotic dialogue? Is the qualifier “semiotic” important in this context?

**Strengths And Weaknesses:**

**Strengths:**

* Extensive analysis of a successful training run and various qualitative aspects of the resulting learned language.
* Two scalability studies: increasing the number of nodes from 5 to 50, and the number of agents from 4 to 7 - both studies show difficulties that arise at scale.
* Extensive discussion and qualitative relation of some of the design principles and the qualitative findings to widely discussed concepts in the philosophy of language and linguistics.

**Weaknesses:**

* Manuscript is very hard to follow, with large portions of vague and ambiguous prose (referring to broad and abstract concepts in the philosophy of language and linguistics that are never fully spelled out formally), numerous mistakes and inconsistencies in the mathematical notation, missing experimental details, and general difficulty for the reader to connect the concrete experiment to the abstract principles referred to in a non-vacuous sense.
* The paper is a case study, but otherwise makes few falsifiable claims and does not formulate well-formed hypotheses. No attempt is made to critically verify the claims and hypotheses and rule out alternative explanations for the observations and the various parts of the agent architecture, loss function, and training protocol are introduced ad-hoc and ablations to justify their introduction are missing. Some of the qualitative claims made are not supported by evidence (see further below).
* From the beginning and throughout, the paper refers to human cognition, linguistic aspects of human language, and neural cognitive faculties - but neither the agent architecture, nor the loss function, or the training protocol are in line with modeling practices in cognitive neuroscience or modern computational linguistics. I do not see any non-trivial and novel conclusions about human- or animal language development that can be taken away from the case study.


**Verdict (novelty, quality, clarity, impact):**

The topic of decentralized acquisition of grounded and efficient language among a population of agents is an interesting and challenging topic. As the paper briefly discusses, simulations with (deep) RL agents have recently become a popular tool to study ideas from computational linguistics empirically. While I appreciate the ambition, I think the current paper falls short on several axes. First, the paper makes almost no falsifiable claims, and does not test well formed hypotheses (in computational linguistics or otherwise) which, together with the biologically/neuroscientifically non-plausible agent architecture, loss functions, and training protocol, means the paper currently does not contribute to the understanding in computational linguistics in a generalizable fashion. Second, the current manuscript needs a major revision for clarity, consistent mathematical notation, and needs proper formalization of all centrally important concepts. Third, the paper does not provide generalizable findings that go beyond the specific observations in the specific case study: the task has not been designed to particularly address any shortcomings of other tasks in the modern literature (other than the vague “allowing significant complexity compared to the existing Lewis signalling games”) so it remains unclear why other researchers should adopt the task as a good testbed; similarly, the agent architecture and training protocol are complex with loads of components that are introduced ad-hoc but not empirically justified via ablations - in order for others to adopt the architecture and training protocol, the minimally required components need to be clear and the qualitative and quantitative impact of all the “bells and whistles” needs to be known. Taking all of this together, I currently vote and argue for rejecting the manuscript at this stage and recommend a major revision of the work since some issues will be non-trivial to address.

---
---

Since I cannot fully rule out that I have misunderstood some of the claims, despite a very careful read of the paper, I will list below the claims and stated contributions from the abstract and sec. 1.1 to clarify why I do not consider some of them supported by sufficient evidence, to facilitate the resolution of any potential misunderstandings on my side.

> “. In this paper, we study minimal yet pertinent aspects of glossogenetics, specifically primal human communication mechanisms, through computational modeling.“

Neither the agent architecture, nor the learning protocol (loss functions and specific use of reward-signals for certain functions) are plausible or compatible with the literature in cognitive science and modern computational linguistics. The paper does not study **human** communication mechanism in any non-vacuous way. Additionally, the **minimal** aspect of studying glossogenetics is currently lacking in the study - it is unclear whether all the complexity in the agent architecture and training protocol is needed.

> “agents with local learning and neural cognitive faculties interact through a series of
dialogues” -

Same as before; this sentence strongly suggests human neural cognitive faculties, or artificial cognitive faculties that are non-trivially related to human neural cognitive faculties. Such a claim is not adequate given the current manuscript.

> “ In our examinations, we observe the emergence of successful and efficient communication among static and dynamic agent populations through consistent learning.“

This claim is mainly justified, though I am not sure what dynamic agent populations precisely refers to (the population of agents in the experiment is fixed, only their roles change randomly).

> “In this paper, we study the emergence of certain coherent properties of a language along with other key factors effecting the language among a multi-agent population.”

I personally do not think that these “key factors” are sufficiently studied - what’s missing is a strong set of empirical ablations and experimental variations that clearly show the qualitative and quantitative impact of the various parts of the architecture and training protocol on these key factors.

> “We develop a game setup allowing significant complexity compared to the existing Lewis signalling games…“

This claim is correct. But at the same time the technical setup of the game (not the agents and training protocol) is quite rudimentary by modern standards and other works in this area. I do not consider the development of the language game environment as a contribution of significant scientific effort and impact (which, to be fair, is not required for TMLR acceptance).

> “Additionally, we explored the notion of interchangeability property in the language”

To be precise, an auxiliary loss to facilitate “interchangeability” (which is only vaguely defined in the paper) was introduced, but the effect of this loss was not ablated and I personally would not call this loss an “**exploration of the notion** of interchangeability property in the language”.

> “In this paper, we also introduce efficient communication through the principle of least effort, where the agents are encouraged to convey information in a way that minimizes complexity or cognitive effort.”

Similar to the previous claim, this is also a bit overstated. In practice this boils down to a L2-regularizer over words (that consist of maximally three tokens), and a low-entropy regularizer over the concept-selection. Biologically “least effort” can mean a whole range of other aspects (which is not discussed in the paper), so this is a strong simplification.

> “Moreover, we study the emergent macro behaviour which materializes through the micro dynamics”

This claim is justified.

---

> ### Author Response · Authors · 2024-05-10
> **Response to Reviewer Xenq**
>
> We thank the reviewer for their valuable comments:
>
> - Neither the agent architecture ...
>
> In this paper, our aim is to create an agent that surpasses the typical scope found in existing literature. When we refer to completeness, we envision an agent capable of multifaceted abilities: speaking, listening, interpreting, conceptualizing, achieving cognitive economy, and engaging in decentralized learning through feedback mechanisms, mirroring processes, and tangible guided responses. This ambition entails designing networks with greater complexity to accommodate these diverse functionalities.
>
> - Same as before ...
>
> When we discuss local learning, we're referring to a decentralized approach to learning. In this context, decentralized learning implies that each  agent  learns and adapts independently based on its local observations and interactions, without requiring a central coordinating authority. Regarding neural cognitive faculties, we're specifically alluding to computational models like artificial neural networks. These models are designed to mimic certain aspects of human cognitive structures, such as learning, pattern recognition, and decision-making processes. By leveraging neural networks, we aim to imbue our systems with cognitive abilities akin to those found in human cognition, allowing them to process information, make decisions, and adapt to changing environments in ways that are more reflective of human-like intelligence.
>
> - This claim is mainly...
>
> As you correctly noted, the term "dynamic" in this context pertains to the fluid nature of roles and knowledge structures within the agents. It implies that these entities are not static but rather subject to continuous evolution, adaptation, and reconfiguration over time. In other words, the dynamics encompass the capacity of agents to transition between different roles and to update their internal knowledge representations in response to changing circumstances or input.
>
> - This claim is correct. But at ...
>
> The guessing games is a very complex setting. Given a vocabulary size $\vert K \vert=m$ and concept size $\vert C \vert=n$, there are $n^m$ different vocabulary mappings possible since there are $m$ possible words for each concept. Now all the $N$ agents have to converge to one of these $n^m$ mappings which further increases the complexity of the learning by a power $N$. This is in addition to the guessing game where guessing game where the associated concept of the uttered word cannot be uniquely identified by the listener and hence can cause a combinatorial explosion.
>
> - To be precise, an ...
>
> In this context, "interchangeability" denotes a state of homogeneity, where an agent possesses the ability to both speak and listen interchangeably. This stands in contrast to the predominant approach found in existing literature, where agents are typically assigned fixed roles, either as speakers or listeners. Achieving this interchangeability is made possible through the design of mirror networks. Here, we conceptualize the speaking module as a mirror image of the listening module, and vice versa. This design ensures that the functionalities required for speaking and listening are inherently similar and interchangeable within the agent's architecture. Consequently, the agent can seamlessly switch between speaking and listening roles as needed, fostering a more versatile and adaptable communication framework.
>
> - Similar to the previous claim ...
>
> When we refer to "least effort" within the domain of human language, we are essentially describing the principle of minimizing the exertion required for communication. This concept is intricately tied to how individuals conceptualize a given topic. In essence, the level of effort required to communicate effectively depends on the complexity and depth of conceptualization. In our approach, we aim to calibrate the network in such a way that it selects all the relevant concepts necessary for expressing a given topic. This ensures that the communication process is streamlined and efficient, with the network automatically prioritizing the most pertinent information.
> Our intuition regarding this approach is supported by the observation of Zipfian characteristics in language. As highlighted in "Human behavior and the principle of least effort: An introduction to human ecology" by George Kingsley Zipf, linguistic patterns often exhibit a distribution where a few concepts or words are used frequently, while many others are used infrequently. This observation suggests that language tends to optimize for minimal effort, aligning with our approach of selecting only the most relevant concepts for communication.

---

> ### Author Response · Authors · 2024-05-10
> **Continuation to previous comment**
>
> # Minor
> - What makes the language game in Fig. 1 “special” ...
>
> There exists various language games depending on the complexity and goals to be accomplished in the game. We chose a guessing game setting which adds complexity to the the Lewis signalling game. Our current setting itself is considerably complex; Given a vocabulary size $\vert K \vert=m$ and concept size $\vert C \vert=n$, there are $n^m$ different vocabulary mappings possible since there are $m$ possible words for each concept. Now all the $N$ agents have to converge to one of these $n^m$ mappings which further increases the complexity of the learning by a power $N$. This is in addition to the guessing game where guessing game where the associated concept of the uttered word cannot be uniquely identified by the listener and hence can cause a combinatorial explosion.
>
> - Problem formulation: what precisely are “coherent symbolic structures”?
>
> Coherent symbolic structures refers to the grounded, shared vocabulary and sentential forms among the agent population.
>
> - Problem formulation: “presenting the ...
>
> Face-to-face here means that agents are directly involved in the communication and are located at the same vertex. There are no intermediaries involved between agents during communication while noise-free communication means the utterances by speaker reaches listener unambiguously. Assumption of noise-free channels in the language settings are very common in the literature. We will definitely update the sentence to make it unambiguous.
>
> - I cannot connect the problem formulation and Fig. 1 to the ...
>
> In this paper, our emergent communication setting is  formalised using a Markovian game framework. In Section $1$, we explain the generalised Markov game framework while subsequent sections delve into the specifics of the language game, elaborating on observation space and action space.
>
> - “The segments and sectors provide a conceptualization ...
>
> When we mention 'cardinal directions,' we are referring to the concepts of North, South, East, and West, which divide space into four regions. In our game setting, agents conceptualize space using sectors and segments. Through our statements, we aim to demonstrate the coherence of our conceptualization by drawing parallels with the notion of cardinal directions. Given that the encoding is pre-fixed as ground truths, permutations are irrelevant in this context.
>
> - Is the concept selection bit-vector really a vector of binary ...
>
> Yes, the concept selection module produces the concept selection vector of binary bits, where $0$ represents not selecting the corresponding concept while $1$ indicates selecting the concept for communicating the target vertex. The L2-norm of the vector 'b' is minimized to reduce the description length, indicating the agent's preference for minimal concept utilization. This process should be interpreted alongside the communication outcome, where rewards are reinforced. The interaction between these components influences parameter adjustments towards achieving a balance between description length and rewards.
>
> - Why is there Gaussian noise on *d* and *w* ...
>
> The noise accompanying  the distance and angle describes the error associated with the partial observability of the agents. These noises are white Gaussian noise,  indicating the agents' inability to measure distances and angles with absolute certainty. This mirrors real-life scenarios where humans can only measure approximately.
>
> - Why is the concept selection module needed? ...
>
> We aim to separate conceptualization and utterance within an agent, and enabling a differentiable channel between them to facilitate mutual influence. Concept selection provides the functionality of efficiently choosing the relevant concepts to be communicated to the listener for the given target node.  The utterance module can only utter discrete words, where the concepts are passed in a sequence, while a separate module allows agent to decide upon the relevant concept sequence to communicate. This flexibility empowers the agents to structure their communication effectively, while also converging on a shared symbol for silence, enhancing the sophistication and adaptability of communication.

---

> > ### Author Response · Authors · 2024-05-10
> > **Continuation to previous comment**
> >
> > - 5.1: It looks like guidance is weakening the notion of decentralized learning ...
> >
> > The speaker guides the listener to the target vertex in the case of failed communication rather than telling the underling concepts (segment, sector, color) used to describe the target vertex. The guidance here implies that the speaker reveals the target node to the listener which is a tangible entity. However, he does not reveal the concepts he used to describe it (which are intangible, hence impossible to show unless telepathy is allowed). Since a node can be conceptualized in multiple ways, the listener could not deduce the concepts underlying the utterance. This we referred to as the poverty of stimulus. The listener addresses the ambiguity by mapping equally  all the concepts associated with the target node to the uttered words. This kind of induction will help the agents to gradually narrow down the mappings as the interactions increases. Here, guidance does not reduce to centralised learning problem. In the centralised learning setting, the speaker and listener network acts as one big network (by making the channel between them differentiable) allowing back-propagation on full network which results in the callibration of the parameters of both speaker and listener. However, in decentralised training the speaker and listener are two standalone networks and the backpropogation in the listener  does not influence the speaker's network.
> >
> > - 5.2: Mirror loss needs an ablation ...
> >
> > Mirror loss synchronises the reverse mapping of the agent. Without mirror loss, interchangeability property cannot be integrated into agent's module. The mirror loss ensures continuity during role switching. With the mirror networks, an agent cannot act as a speaker and a listener. This is the notion of homogenity we were addressing in the paper. Mirror loss enables an agent to learn while speaking and listening. We believe that what the speaker speaks will also influence him while interpreting when he is a listener. The mirror loss convergence can be observed in figure 11.
> >
> > - 5.2: Sentence length is only one factor related to the ...
> >
> > It is not really the sentence length, rather the minimum concepts required to convey the target. This can be read with respect to the information theory, where the objective is to transfer information with minimum number of bits. There exists a discrete communication channel between speaker and listener where the speaker can only utter words (chosen from the given vocabulary) for the concepts provided by the concept selection module. Hence, deciding upon the essential concepts to utters is considered to minimise the effort in communication. The key point to be noted here is that the communication channel is discrete and we are not considering a machine to machine communication.
> >
> > - 5.5: There is a mistake in the signs of the loss terms ...
> >
> > We totally agree with the reviewer. It is a typo error which we will definitely update.
> >
> > - Ideally show that results are qualitatively the same ...
> >
> > We have acknowledged in the section $6.2$ that the game converges almost ($\approx 95\%$) times and the occasional failures are due to the lack of positive rewards which arises due to random initialization of the neural network weights and the distribution bias of the source, topic vertices pair chosen for the conversations. This observation resonates with real-world scenarios where multiple timelines are possible. Here, we present the most significant evolutionary paths observed in our simulations.
> >
> > - 6.3.2 Emergence of a word for silence: the text suggests a surprising emergence of a shared word for silence, while this is actually forced via a significant auxiliary loss (Eq. 7) that rewards individual agents to reuse the same word for silence (across concepts) and agents are forced to coordinate their vocabularies globally (and use short sentences).
> >
> > The reward for the null concept is integrated to introduce a bias for the speaker to use the same word for null concepts which is similar to humans trying to use same word used earlier. There exist no telepathy between speaker and listener to use same word any concept and a successful mapping could only emerge through feedback mechanism.
> >
> >
> > - 6.5.1 This is an observation for a single training run - what generalizable finding can be taken away?
> >
> > In our experiments we show successful convergence of language for active concepts during the conversation. The key observation here is the emergence of a grounded vocabulary among agents along the other emerged language behaviour.

---

> > > ### Comment · Reviewer_Xenq · 2024-05-14
> > > **Thank you for the detailed response**
> > >
> > > I want to thank the authors for the detailed response.
> > > Most answers are in line with my understanding after originally reading the paper, but unfortunately do not address the issues raised or add substantial new information in most cases. Since there are no open clarifying questions from the authors, I believe any potential misunderstanding on either side has been resolved now.
> > >
> > > In light of this updated information, most of the criticism in my review remains, and I currently stand by my original assessment and recommendation.
> > >
> > >
> > > Detailed comments:
> > >  * Thanks for clarifying that the concept-selection vector is actually a binary vector.
> > >  * Re mirror loss: When I asked for the ablation, I was curious to see whether the loss is actually needed or not for training to converge. I currently believe that it helps speed up training but don't think that it is crucially needed. The same argument goes for many other parts of the architecture. Additionally, I do not believe that the mirror loss in its current form is neuroscientifically plausible.
> > >  * Re face-to-face: since the channel is already defined to be noiseless, I do not think that the qualifier "face-to-face" adds significant information.
> > >  * Complexity of the game: I agree that there are many game-states (including agent's vocabularies for speaker and listener) and that a brute-force-search would be futile for at scale. But I do not consider the game a particularly hard learning problem by modern ML standards.
> > >  * Re principle of minimal effort: sentence length is at best a crude approximation to "minimal effort" in human/biological language, where an array of other factors plays a role in terms of "effort". It is fine to use a sentence-length regularizer in the paper, but its biological inspiration/implications are a bit overstated.
> > >  * Re results qualitatively the same: yes, I understood that the paper states that 95% of training runs converge. But the question is do the same qualitative aspects (Zipf's law, shared word for silence, etc.) reliable emerge in these training runs?
> > >  * Re word for silence: why is the principle of least effort (sentence length regularizer) not enough? Is the extra loss term crucially needed (which can only be shown empirically by removing it and demonstrating that no shared word for silence emerges)?

---

> ### Author Response · Authors · 2024-05-25
> **Response**
>
> We thank the reviewer for the review.
>
> The aim of this paper is to construct a language game environment enabling agents to both communicate and comprehend. We introduce a framework featuring explicit concept and vocabulary spaces for the agents, with the goal of fostering a shared language among independently calibrated agents, thereby naturally establishing correlation between them without explicitly correlating them. This approach diverges from existing works such as "Emergence of Grounded Compositional Language in Multi-Agent Populations," which focuses on generating a compositional language using non-homogeneous agents with fixed roles through a differentiable channel between them. Furthermore, it's essential to note that all rewards within our framework are tangible, meaning that both the speaker and the listener can verify the outcomes of their actions and evaluate them. These rewards stem directly from their evaluations, without the involvement of any central agency. This aspect mirrors human behavior, where individuals assess the consequences of their actions and receive feedback based on their evaluations without external intervention.
>
> Central to our approach is the concept of mirror networks. The utterance network is characterized as a conditional probability distribution over vocabulary space conditioned on the concept, while the listening module is also a conditional probability distribution over concept space conditioned on the vocabulary. Thus, these networks are viewed as mirror counterparts, reflecting the symbiotic relationship between speaking and listening. In real scenarios, knowledge gained through speaking informs listening and vice versa, necessitating synchronization rather than independence between the networks, which is the rationale behind the mirror loss.
>
> Regarding concept selection, it is hypothesized that humans optimize their effort, which we characterize in the agent's perception through the minimization of description length. This optimization drives the language to exhibit Zipfian characteristics observed across human languages. However, it's important to note that the final objective is non-convex, thus not all components can achieve optimality. The emergent language arises through trial and error and independent reinforcements within the gameplay dynamics, making it complex to predict the outcome or nature of the emergent language.
>
> It's not guaranteed that a language will emerge in every gameplay scenario, as evidenced by some evolutionary trajectories lacking emergent behavior. This is being reported in the paper as we observe that some trajectories do not exhibit any structured and shared communication behaviour. This alludes to the fact that the game is challenging as we stretch the concept space further.  Additionally, we address the notion of silence, which is learned by the agents rather than pre-fixed as in existing works. By rewarding consistent usage of the same word for silence, akin to human communication patterns, we incentivize communication coherence.
>
> The reviewer requests further studies, but the analysis already covers how various components influence the emergent language and its characteristics. Comparing with other models from existing literature isn't feasible, as the objectives and models differ, and our focus isn't on commenting about neural models which are studied in neuroscience. Our neural architectures adhere to standard RNN models without specific modifications to achieve emergent behavior.
>
>
> The core of our  study lies in the complexity arising from the interplay dynamics, coupled with the guessing game nature and decentralized learning. Despite this complexity, our study demonstrates the emergence of a language with key characteristics such as groundedness, shared understanding, coherence, compositionality, word order, and Zipfian distribution. Making direct comparisons between our findings and real-world human language settings is indeed challenging and somewhat unfair given the immense complexity inherent in human communication. However, it remains crucial to assess our results within the framework of existing artificial language emergence settings, which we extensively review in our introduction section.
>
> P.S:. I kindly request the reviewer to highlight any errors found in our mathematical derivations, as these comments have the potential to impact the credibility of our approach. Additionally, I would like to clarify that "face-to-face" simply implies that both the listener and the speaker are located at the same node. Furthermore, it's important to note that noise is intentionally added to ensure that the distance and angle considered by the agents are only estimates of their true values. This approach is crucial for accurately simulating real-world scenarios where agents may have imperfect information or encounter environmental variability.

---

> > ### Comment · Reviewer_Xenq · 2024-05-26
> > **Thank you for the additional comments**
> >
> > Thank you for repeating the rationale behind many of the design decisions, and highlighting some relevant results in the manuscript. The reasoning is coherent with my initial reading of the paper (some of which I have criticised in my original review). I appreciate the complexity of the various parts and factors that went into the case-study, and have no doubt that a fair amount of work went into getting all the bits and pieces right, and work well together. To clarify: I did not expect a direct experimental comparison of other models in the literature, but a motivation of all parts of the current model by referring to standard approaches and techniques and pointing out why they are not sufficient and how the current paper improves upon them. "The reviewer requests further studies, but the analysis already covers how various components influence the emergent language and its characteristics" - to me personally, not all claims are supported by sufficient evidence; there is some analysis but it does not cover all parts of the model and training procedure (see my orig. review where I discuss each claim in detail).
> >
> > Re: mathematical errors - see my orig. review for an extensive list of small errors and inconsistencies in notation.

---

### Decision · Action_Editor_KXMd · 2024-09-13

**Recommendation:** Reject

**Comment:**

The paper presents a study of emergent communication of LSTM-based agents participating in a guessing game setting that involves communicating the location and color of different vertices of a graph. The paper describes neural network implementations of the speakers and listeners that, in conjunction with a specific training setup (e.g., different time scales, several auxiliary losses) result in communication that is grounded and compositional.

As noted above, the question of what minimum characteristics are necessary to realize human-like communication that is compositional and grounded is interesting. However, the reviewers are in agreement regarding the significant weaknesses with the paper as currently written. Among them, the paper makes a number of strong claims that are not falsifiable rather than well formed hypotheses that can then be verified in a principled manner. The reviewers find the study to be unnecessarily complex and monolithic, involving a large number of variables/components and many design decisions many of which are ad-hoc and different from existing work, which, together with the absence of ablations, make it impossible to get a sense of which aspects are responsible for the different phenomena that are observed. Related, the niche nature of the system and task make it unclear how the findings can be generalized beyond the particular case study considered here. Thus, the reviewers agree that it is not obvious what readers are supposed to take away from the paper.

**Audience:**

The high-level topic of the paper---investigating the properties of communication necessary for human language to arise---is of interest to many in the community.

**Claims And Evidence:**

The reviewers agree that the paper makes a number of claims whose correctness is difficult to evaluate. As noted by Reviewer 2sqP, some of the claims regarding language are arguably fundamentally wrong, such as the claim that language is unambiguous. The reviewers emphasize that, as part of a significant revision, the paper should formulate claims as falsifiable hypotheses and describe an effective strategy for evaluating their correctness.